# Flotillins promote T cell receptor sorting through a fast Rab5–Rab11 endocytic recycling axis

Gregory M.I. Redpath[1], Manuela Ecker[1], Natasha Kapoor-Kaushik[1], Haig Vartoukian[1], Michael Carnell[2], Daryan Kempe[1], Maté Biro [1], Nicholas Ariotti[3,4] & Jérémie Rossy [1,5]*

The targeted endocytic recycling of the T cell receptor (TCR) to the immunological synapse is essential for T cell activation. Despite this, the mechanisms that underlie the sorting of internalised receptors into recycling endosomes remain poorly understood. To build a comprehensive picture of TCR recycling during T cell activation, we developed a suite of new imaging and quantification tools centred on photoactivation of fluorescent proteins. We show that the membrane-organising proteins, flotillin-1 and -2, are required for TCR to reach Rab5-positive endosomes immediately after endocytosis and for transfer from Rab5- to Rab11a-positive compartments. We further observe that after sorting into in Rab11a-positive vesicles, TCR recycles to the plasma membrane independent of flotillin expression. Our data suggest a mechanism whereby flotillins delineate a fast Rab5-Rab11a endocytic recycling axis and functionally contribute to regulate the spatial organisation of these endosomes.

[1] EMBL Australia Node in Single Molecule Science, School of Medical Sciences and the ARC Centre of Excellence in Advanced Molecular Imaging, University of New South Wales, High St Gate 9, 2052 NSW Sydney, Australia. [2] Biomedical Imaging Facility, Mark Wainwright Analytical Centre, University of New South Wales, High St Gate 9, 2052 NSW Sydney, Australia. [3] Electron Microscopy Unit, Mark Wainwright Analytical Centre, University of New South Wales, 2052 NSW Sydney, Australia. [4] Department of Pathology, School of Medical Sciences, University of New South Wales, 2052 Sydney, NSW, Australia. [5] Biotechnology Institute Thurgau at the University of Konstanz, 8280 Kreuzlingen, Switzerland. *email: jeremie.rossy@bitg.ch

Endocytosis removes proteins and lipids from the plasma membrane, while recycling continuously returns those same proteins and lipids back to the cell surface. This cycle between endocytic uptake and recycling was first envisaged only as a mechanism to separate receptors from their ligands before returning them—or not—to the plasma membrane for further engagement. It appears however that the picture is more complex and that endocytic trafficking governs the balance between what is stored intracellularly and what is made available for interaction at the cell surface[1,2]. Hence, fine tuning of endocytic recycling constantly remodels the local density and spatial distribution of surface proteins in response to specific cellular processes. Targeted recycling is, for instance, essential to establish and maintain functional polarity by spatially restricting membrane proteins at specific localisations in the cell. As such, the endocytic recycling system plays a central role in a multitude of cell types and processes[1–3]. This is especially true for T cells whose activation relies on endocytosis and recycling of T cell signalling proteins to and from the immunological synapse[4–14].

A critical yet still poorly understood step in endocytic trafficking is sorting at the Rab5-positive early endosomes[2]. Multiple sorting mechanisms determine if endocytosed proteins are sent to the lysosome or returned to the plasma membrane via dedicated recycling endosomal compartments demarked by Rab4 or Rab11[15,16]. Sorting into Rab4-positive and Rab11-positive recycling endosomes is thought to occur via tubulation and vesicle fission from Rab5-positive endosomes, as opposed to endosomal maturation leading to late endosomal/lysosomal compartments[17,18]. Microtubule motors play an essential role in tubulation and fission by powering elongation of membranes along microtubules[19,20]. They also play a fundamental role in endocytic sorting by mediating the spatial organisation of endosomes. Dynein, for instance, is required for the sorting of transferrin (Tf) and its receptor (TfR) for recycling by powering the movement of endosomes towards perinuclear recycling compartments[21]. This movement is coupled to membrane organisation and tubulation by the membrane-binding protein, sorting nexin 4 (SNX4)[22]. More generally, coupling between membrane domains delineated by sorting nexins and microtubule motors maintains the topography of sorting endosomes[23]. By acting as entry doors for distinct vesicular pathways, membrane domains can also contribute to endocytic sorting at the plasma membrane before entry into early endosomes[24,25]. Similarly, distinct membrane domains can maintain cargo segregation upon exit from early endosomes[26]. Hence, preserving the identity of distinct endosomes, or of membrane domains within endosomes, is central to endocytic sorting either by (a) maintaining cargo segregation while endocytosed proteins are travelling through successive endosomes or (b) by coupling distinct endosomes to specific microtubule motors.

In T cells, various proteins have been shown to contribute to the recycling of TCR[14,27], but no mechanism has been put forward as to how TCR is so efficiently sorted for recycling back to the immunological synapse. The cytoplasm of T cells is constricted and makes an unfavourable environment for the formation of endocytic tubules. It is therefore very likely that membrane domains within endosomes and their relative spatial organisation regulate how TCR is sorted for recycling. Flotillins, otherwise known as Reggies, are membrane-binding proteins that oligomerise to shape membrane domains, including in T cells[28–32]. The membrane-organising properties of flotillins are thought to be especially relevant in the context of endosomal sorting and endocytic recycling[33]. Flotillins have been implicated in receptor and integrin recycling via SNX4 and Rab11a-positive recycling endosomes[34–36]. Flotillins also interact with components of the cytoskeleton[37,38]. Notably, flotillin-rich microdomains have been shown to cluster and promote activation of dynein on endosomes to direct their transport along microtubules[39].

Here we show that flotillins are required to sort internalised TCR into a fast Rab5-Ra11 endocytic recycling pathway. To fully understand each and every step of this process, we establish a live-cell imaging tool kit including single and two-photon photoactivation of fluorescent proteins, TIRF microscopy and optogenetics. Using this approach, we show that less TCR reaches Rab5-positive endosomes in flotillin KO cells, and that flotillins play an even larger role in sorting TCR from Rab5 to Rab11-positive endosomes. Once incorporated into Rab11a vesicles however, TCR recycles to the immunological synapse independently of flotillins. Finally, our data suggest that flotillins (a) define an endocytic fast lane to get TCR into Rab11-positive endosomes and (b) contribute to spatially organise Rab5 and Rab11-positive compartments for efficient recycling to occur. Altogether, our data indicate that flotillins organise membrane domains supporting Rab11-mediated TCR recycling in activated T cells.

## Results

**Flotillins mediate TCR sorting into Rab5 and Rab11 endosomes.** Flotillins have been implicated in recycling of multiple cell surface proteins, including TCRζ[14,34,35], while we have previously shown that they do not contribute to TCRζ endocytosis[14]. The precise role flotillins play in endocytic recycling remains to be elucidated. The ability of flotillins to structure membrane domains led us to hypothesise that they may not promote recycling directly, but rather organise the sorting of internalised cargoes into recycling pathways.

It is challenging to identify the target compartments of endocytosed receptors. To visualise in real-time the endosomal destination of internalised TCRζ, restricted areas of the plasma membrane of cells expressing TCRζ-PAmCherry and GFP-Rab4, GFP-Rab5 or GFP-Rab11a were repetitively illuminated with 405 nm light to trigger localised photoactivation of PA-mCherry[14]. The GFP signal was used to produce a mask defining the endosomes of interest. The PA-mCherry fluorescence intensity within this mask was measured to quantify the incorporation of endocytosed TCRζ into Rab4, Rab5 or Rab11a compartments (Fig. 1a, Supplementary Movie 1).

TCRζ rapidly entered Rab4 and Rab5 endosomes after endocytosis (Fig. 1b, c, e, f, Supplementary Movie 1). We tested the statistical difference of the TCRζ-PAmCherry signal in these endosomes at every time point between WT and flotillin1/2 double knockout Jurkat T cells (FlotKO cells hereafter). TCRζ intensity within Rab4-positive endosomes never diverged between WT and FlotKO cells (Fig. 1b, c). Fitting a linear regression curve to the TCRζ signal detected in the mask revealed that there was no difference in the amount of internalised TCRζ leaving Rab4-positive endosomes between WT and FlotKO over time (Fig. 1d). In contrast, we observed less internalised TCRζ reaching Rab5-positive compartments in FlotKO compared to WT cells from 7.5 to 87.5 s after endocytosis (Fig. 1f, dashed line to solid line), indicating TCRζ entry into Rab5 endosomes is impaired shortly after endocytosis in FlotKO cells. Furthermore, linear regression indicated TCRζ-exited Rab5 endosomes at a significantly slower rate in FlotKO compared to WT cells (Fig. 1g). We confirmed these results in a second flotillin knockout Jurkat T cell line (Supplementary Fig. 1a), finding TCRζ entry into (Supplementary Fig. 1b) and exit out of (Supplementary Fig. 1c) Rab5 endosomes was also significantly reduced in the second FlotKO clone. These data indicate that flotillins regulate both TCRζ entry into, and exit from, Rab5-positive endosomes.

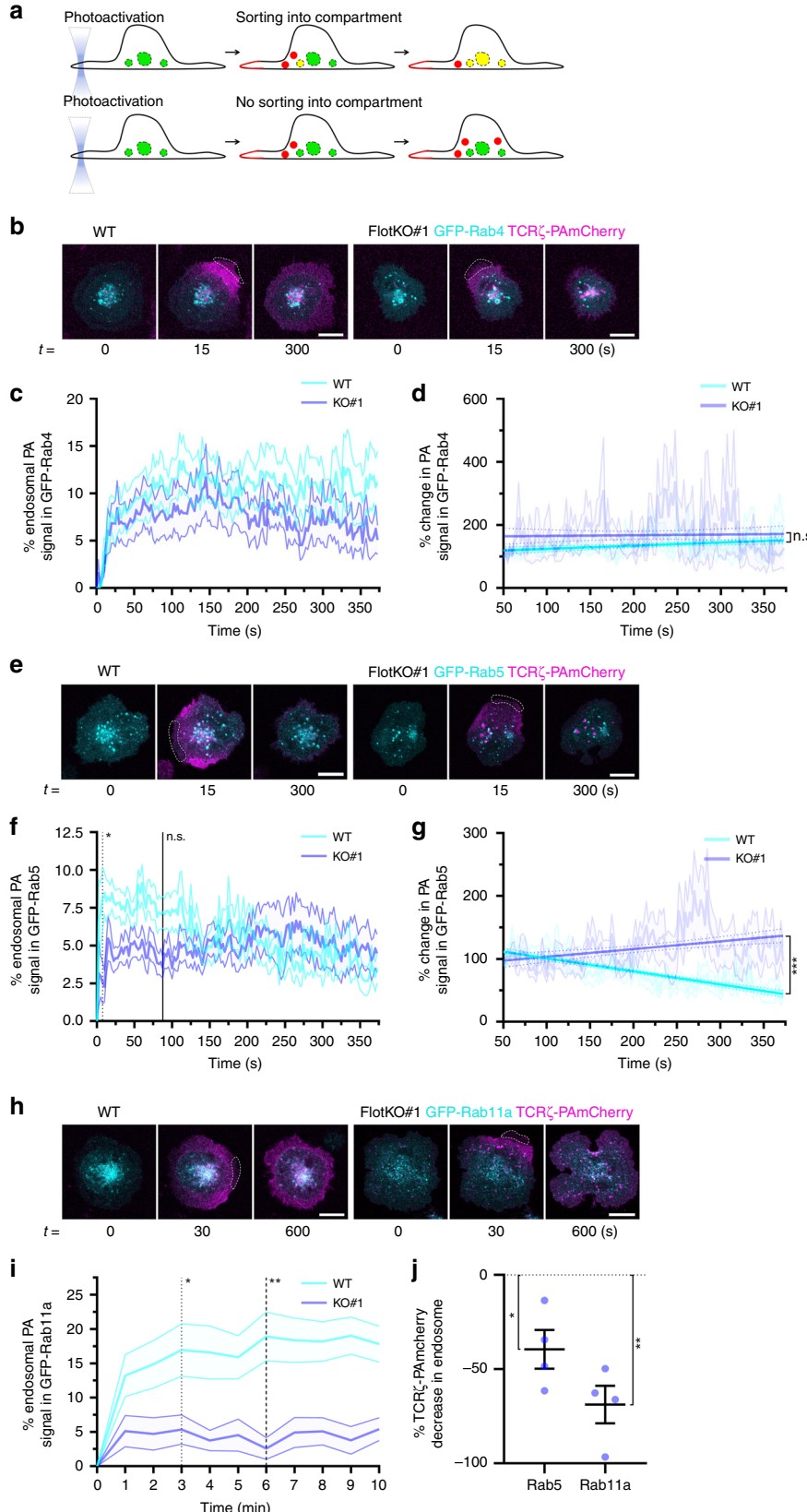

Endocytosed TCRζ entered Rab11a endosomes within 1-min following internalisation (Fig. 1h, i, Supplementary Movie 1). From 3 min onwards, there was a lower amount of TCRζ-PAmCherry being incorporated into Rab11a endosomes in FlotKO compared to WT cells (Fig. 1i, dotted line). Hence,

incorporation of TCRζ into both Rab5 and Rab11a endosomes was significantly reduced in FlotKO cells when compared to WT cells (Fig. 1j, $t = 7.5–87.5$ s for Rab5, $t = 6–9$ min for Rab11a, Supplementary Movie 1). However, the reduction due to flotillin knock-out in the amount of TCR being incorporated into Rab11a

**Fig. 1** Flotillins mediate sorting of internalised TCRζ into Rab5 and Rab11a endosomes. **a** Schematic depicting photoactivation to investigate sorting. Top: sorting of photoactivated endosomes (red) into the Rab compartment (green, yellow = merged), bottom: photoactivated endosomes not sorting into the Rab compartment. Dashed lines: quantified compartment mask. **b** Representative images of photoactivated (dashed region) TCRζ-PAmCherry sorting into EGFP-Rab4 in WT (left) and FlotKO (right) Jurkat T cells. Cells photoactivated for five frames with 2.5 s intervals, then imaged every 2.5 s for 150 frames. **c** % endosomal TCRζ-PAmCherry intensity in Rab4 in WT or FlotKO cells. Dashed line represents timepoint where TCRζ-PAmCherry intensity significantly diverges between WT and FlotKO. **d** Percentage reduction in endosomal TCRζ-PAmCherry intensity in Rab4 from the 50 to 55 s timepoint. Fitted linear regression line is bolded. n.s. indicates slopes are not significantly different. **e** Representative images of photoactivated (dashed regions) TCRζ-PAmCherry sorting into EGFP-Rab5 in WT (left) and FlotKO (right) cells. Cells imaged as in **b**. **f** Percentage endosomal TCRζ-PAmCherry in Rab5 in WT or FlotKO cells over time. **g** Percentage reduction in endosomal TCRζ-PAmCherry intensity in Rab5 from 50 to 55 s. Fitted linear regression line is bolded. ***Indicates slopes are highly significantly different. **h** Representative images of photoactivated (dashed region) TCRζ-PAmCherry sorting into EGFP-Rab11a in WT (left) and FlotKO (right) cells. Cells were photoactivated for five frames with 2.5 s intervals, then imaged at 1 frame per minute for 10 min. **i** % endosomal TCRζ-PAmCherry in Rab11 in WT or FlotKO cells over time. **j** Percentage of TCRζ-PAmCherry intensity in Rab5 and Rab11 in FlotKO cells compared to WT. *$p < 0.05$, **$p < 0.01$ from one sample $t$-test comparing values to normalised WT mean of 100% **j**. Dotted line = majority of timepoints after which $p < 0.05$, dashed line = $p < 0.01$, solid line = timepoints after which $p > 0.05$ from Wilcoxon rank sum test. Data points = means of $n = 3$ (Rab4, WT: 5; 4; 5 cells and FlotKO: 4; 3; 4 cells per experiment), $n = 4$ (Rab5, WT: 7; 5; 6; 7 cells and FlotKO: 3; 7; 4; 3 cells per experiment) and $n = 4$ (Rab11, WT: 6; 1; 7; 4 cells and FlotKO: 4; 4; 2; 2 cells per experiment) biologically independent experiments, error bars = mean ± SEM. Scale bar = 5 μm

endosomes was larger than in Rab5 endosomes (68.8 ± 19.9% and 39.5 ± 20.9% respectively). In conclusion, these experiments show that flotillins appear to mediate sorting along the endocytic recycling pathway at two points with varying degrees: (i) primarily by mediating TCRζ transition between Rab5 sorting and Rab11a recycling endosomes, and to a lesser extent (ii) by supporting TCRζ entry into Rab5-positive sorting endosomes.

**TCR recycles from a flotillin and Rab11-positive compartment.**
As flotillins appear to mediate sorting at multiple points along the endocytic recycling pathway (Fig. 1), we sought to visualise the recycling of TCRζ to further dissect how flotillins facilitate TCRζ return to the plasma membrane. To do so, we established a microscopy approach to specifically identify the components of the endocytic machinery responsible for TCRζ return to the plasma membrane. We used a two-photon laser to selectively photoactivate (or photoswitch, referred to as photoactivation hereafter) and image TCRζ labelled with the photoswitchable fluorescent protein PS-CFP2 (TCRζ-PSCFP2) in various recycling compartments (Fig. 2a). Two-photon illumination at 800 nm, 2.5 μm inside the cell resulted in robust photoactivation of intracellular TCRζ-PSCFP2 without any detectable plasma membrane signal in fixed cells (Supplementary Fig. 2a). In comparison, conventional 405 nm laser illumination 2.5 μm inside the cell resulted in large amounts of photoactivated TCRζ-PSCFP2 signal at the plasma membrane above and below the focal plane (Supplementary Fig. 2b), highlighting the specificity of the two-photon approach.

To quantify recycling in live cells, we measured the fluorescence intensity at the plasma membrane after two-photon photoactivation of intracellular TCRζ-PSCFP2. To establish a baseline, TCRζ-PSCFP2 was photoactivated inside TCRζ-mCherry-positive endosomes 2.5 μm deep at the centre of the cell (Fig. 2b, dashed circle). We observed that TCRζ recycled from these compartments, with increasing intensity of photoactivated TCRζ-PSCFP2 measured over time at the plasma membrane (Fig. 2b–d, Supplementary Movie 2, top left). Consistent with our previous report that TCRζ recycling is impaired in FlotKO cells[14], TCRζ-PSCFP2 levels at the plasma membrane were significantly reduced in FlotKO cells compared to WT (Fig. 2c, d, Supplementary Fig. 2c). As negative control, photoactivation of empty cytoplasm 2.5 μm within the cell resulted in no quantifiable photoactivated PSCFP2 signal at the plasma membrane (Fig. 2c, d, Supplementary Movie 2, top right). Of note, the mean photoactivated signal of TCRζ-PSCFP2 in WT and FlotKO cells was similar (Fig. 2e), confirming that the impaired recycling in FlotKO cells observed in this assay was due

to a bona fide recycling defect and not to a less efficient photoactivation in FlotKO cells.

We then photoactivated TCRζ-PSCFP2 within the flotillin-positive endosome cluster at the cell centre. We co-expressed flotillin-1 and -2 in equal quantity to minimise potential effects of flotillin over-expression and ensure equal flotillin quantities were present for flotillin microdomain formation[32,40]. Hereafter, every case of flotillin expression refers to expression of both flotillin-1 and flotillin-2 and is termed flotillin1/2. We observed similar levels of TCRζ-PSCFP2 recycling when it was photoactivated within central flotillin1/2-positive compartments compared to compartments labelled with TCRζ (Fig. 2f, g, Supplementary Movie 2, bottom left). However, a significant increase in TCRζ recycling compared to the baseline was observed when TCRζ-PSCFP2 was photoactivated in Rab11a-positive compartments at the cell centre (Fig. 2f, g, Supplementary Movie 2, bottom right). These results confirm that both flotillins and Rab11a label the TCRζ recycling compartment. They further indicate that TCRζ contained in Rab11a-positive endosomes is especially likely to reach the plasma membrane.

We then performed a nearest-neighbour analysis to determine if the vesicles positive for photoactivated TCRζ-PSCFP2 outside of the photoactivation region contained flotillin1/2 or Rab11a. The percentage of vesicles leaving the central endosomal compartments that were positive for flotillin1/2 (11.8 ± 2%) was not significantly different compared to a randomised control (8 ± 1%, Fig. 2h). In contrast, a significantly higher percentage of TCRζ-positive vesicles leaving these compartments contained Rab11a (26.7 ± 2.6%, Fig. 2h). These results suggest that vesicles that transport TCRζ from intracellular compartments to the plasma membrane do not contain flotillin and are positive for Rab11a. They further suggest that flotillins contribute to sorting TCRζ prior to entry into Rab11a-positive endosomes.

Finally, combining the results from the sorting experiments (Fig. 1) with the data obtained using our two-photon recycling assay allowed us to estimate the time taken for TCRζ recycling to occur. The half-maximal time for TCRζ sorting into Rab11a-positive endosomes was 186.6 ± 93.5 s (3 min 6 s, Fig. 1), while the half-maximal for TCRζ to recycle from Rab11a endosomes to the plasma membrane was 375 ± 195.1 s (6 min 15 s, Fig. 2), giving a total time of half-maximal recycling of ~9 min and 21 s. These times are consistent with the fastest observed times for transferrin recycling (half-maximal times of 3 min for internalisation, 5.5 min from internalisation to recycling, total 8.5 min)[41], indicating that flotillin mediates sorting along a fast, Rab11a-dependent recycling pathway.

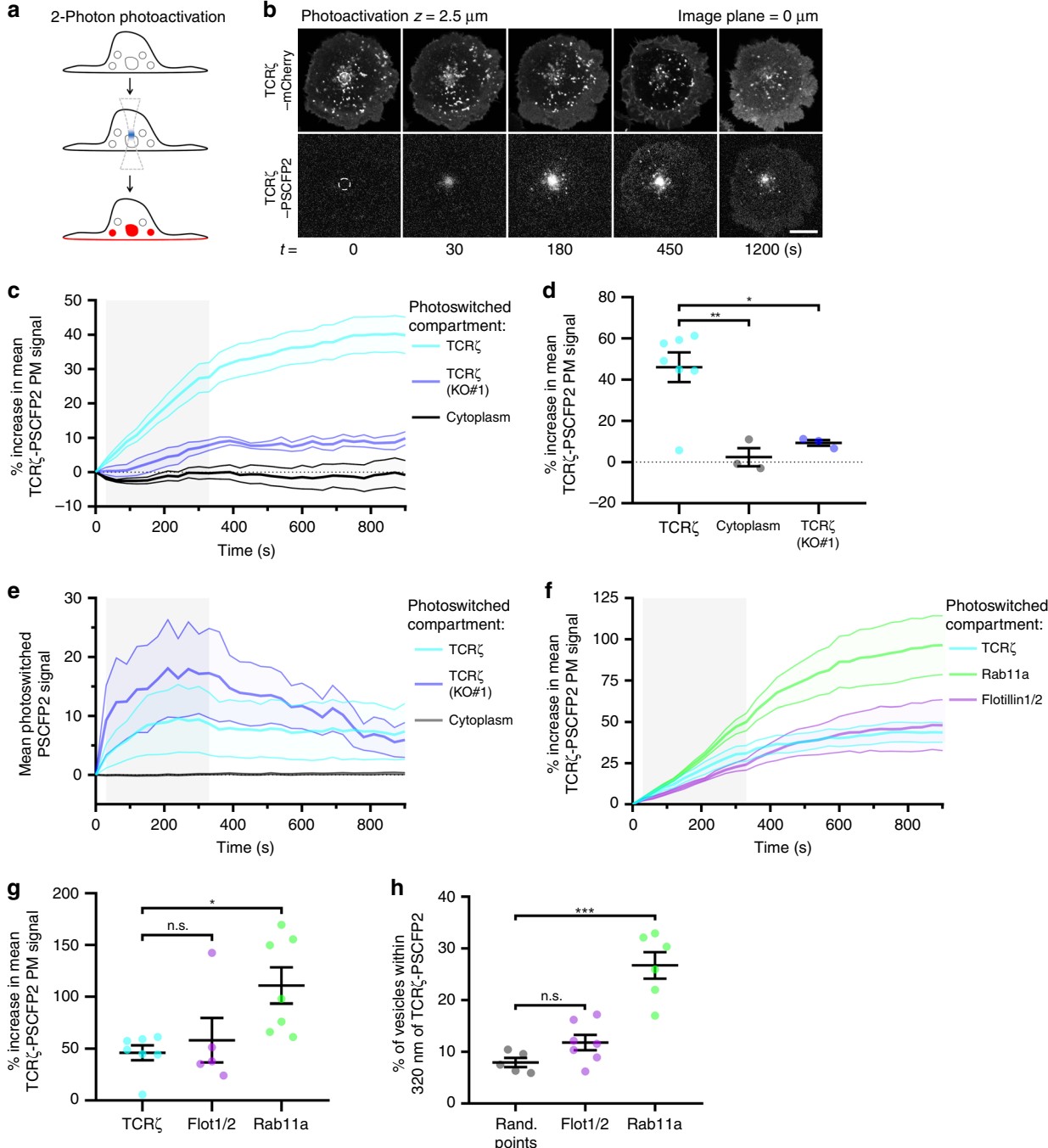

**Fig. 2** TCRζ recycles from an intracellular compartment labelled by flotillins and Rab11a. **a** Schematic depicting two-photon photoactivation to measure recycling from intracellular compartments. **b** Representative confocal images of TCRζ-PSCFP2 photoactivated with 800 nm light from a two-photon laser, 2.5 μm within the cell at the cell centre-localised TCRζ-mCherry compartment displaying recycling to the plasma membrane over time. Dashed circle = photoactivation region. **c** % increase in photoactivated TCRζ-PSCFP2 signal over time at the plasma membrane when activated from cell centre compartments labelled by TCRζ-mCherry in WT or FlotKO Jurkat T cells or cytoplasmic photoactivation. Greyed region = time of photoactivation. **d** Mean TCRζ-PSCFP2 signal from 900 to 1200 s timepoints when photoactivated from the indicated compartments. **e** Mean intenisty of photoactivated TCRζ-PSCFP2 2 μm within the cell from TCRζ-mCherry in WT or FlotKO cells. Greyed region = time of photoactivation. **f** % increase in photoactivated TCRζ-PSCFP2 signal over time at the plasma membrane when activated from cell centre compartments labelled by mCherry-Rab11a, flotillin1/2-mCherry or TCRζ-mCherry in WT Jurkat T cells. Grey box = time of photoactivation. Greyed region = time of photoactivation. **g** Mean TCRζ-PSCFP2 signal from 900 to 1200 s timepoints when photoactivated from the indicated intracellular compartments. **h** Percentage of photoactivated TCRζ-PSCFP2 vesicles within 320 nm of vesicles of the indicated marker outside of the initial photoactivation region. Data points = means of independent experiments, error bars = mean ± SEM. n.s. = not significant, *$p < 0.05$, **$p < 0.01$, ***$p < 0.001$ from a two-tailed unpaired Student's $T$-test of means of $n = 7$ (TCRζ: 3; 3; 1; 1; 1; 1; 1 cells per experiment), 5 (flotillin1/2: 2; 1; 1; 1; 1 cells per experiment), 7 (Rab11a: 1; 1; 2; 1; 1; 1; 1 cells per experiment), 3 (cytoplasm: 3; 1; 2 cells per experiment), and 3 (TCRζ in FlotKO: 1; 1; 2 cells per experiment) biologically independent experiments. Scale bar = 5 μm

**Flotillins mediate TCR sorting into recycling Rab11 vesicles**. Our results indicate flotillins mediate sorting along the Rab5 to Rab11a recycling endosomal pathway (Fig. 1), playing a large role in mediating TCRζ entry into Rab11a vesicles for the return to the plasma membrane (Fig. 2). To further confirm that flotillins exert their effect on recycling by sorting TCRζ into Rab11a vesicles as opposed to directly regulating the Rab11a-dependent recycling process, we used TIRF microscopy to visualise the fusion of TCRζ-positive vesicles with the plasma membrane.

Fusion events of TCRζ-GFP vesicles were imaged in live Jurkat T cells between 5 and 30 min after activation on glass coverslips coated with activating antibodies (Fig. 3a, Supplementary Movie 3, left). Visual identification of fusion events in a blinded analysis revealed a clear reduction in TCRζ-GFP vesicle fusion in two FlotKO cell lines compared to WT (Fig. 3b). We next co-transfected TCRζ-GFP with flotillin1/2-mCherry or mCherry-Rab11a to determine if fusing TCRζ-positive vesicles contained flotillins and/or Rab11a. While flotillin1/2 was present in only $15 \pm 1\%$ of TCRζ-GFP vesicle fusion events with the plasma membrane (Supplementary Movie 3, right), $80 \pm 8\%$ of TCRζ-GFP fusion events involved Rab11a (Fig. 3c, d).

We then quantified all fusion events of GFP-Rab11a in WT and FlotKO Jurkat T cell lines (Fig. 3e, Supplementary Movie 3, bottom left). No significant difference was observed in the number of GFP-Rab11a vesicle fusion events per minute between WT and two FlotKO cell lines (Fig. 3f). This result confirms that flotillins do not play a direct role in Rab11a-mediated vesicle fusion with the plasma membrane. Analysis of mCherry-Rab11a and TCRζ-GFP co-fusion events revealed two populations of fusing TCRζ vesicles in FlotKO cell lines: TCRζ vesicles fusing with Rab11a (as in WT cells), and TCRζ vesicles fusing independently of Rab11a (Fig. 3g, Movie 3, bottom right). Only $40 \pm 8\%$ (FlotKO#1) and $35 \pm 1\%$ (FlotKO#2) of TCRζ fusion events contained Rab11a in FlotKO cell lines compared to $80 \pm 8\%$ in WT cells (Fig. 3f). Altogether, these data indicate that flotillins do not directly regulate the Rab11a-dependent recycling of TCRζ, but rather mediate the efficient transfer of TCRζ into Rab11a vesicles for subsequent re-delivery to the plasma membrane.

**Flotillins localise to Rab5 and Rab11a endosomes after endocytosis**. Flotillins are established membrane domain proteins[33,42,43] that interact with Rab5[44] and Rab11a[34,35]. We therefore hypothesised that they might regulate TCRζ sorting by forming entry doors from the plasma membrane into Rab5 endosomes and subsequently from Rab5 to Rab11a endosomes. To test this hypothesis, we expressed flotillin-2-PAmCherry together with untagged flotillin-1 and GFP-Rab4, 5 or 11a. As for TCRζ in Fig. 1, we visualised how flotillin-2-PAmCherry endocytic vesicles interacted with GFP-Rab compartments and quantified the percentage of photoactivated signal present in these endosomes. If flotillins are indeed an entry door into Rab5 and Rab11a compartments, we would expect to observe endocytosed flotillins first co-occurring with, and eventually being incorporated within, Rab5 and Rab11a-positive endosomes.

No obvious interaction was observed between flotillin-2 endocytic vesicles and Rab4 (Fig. 4a, Supplementary Movie 5, left). Short and transient interactions were evident between flotillin-2 and Rab5 (Fig. 4b, Supplementary Movie 5, middle) while photoactivated flotillin-2 appeared to accumulate in Rab11a compartments over time (Fig. 4c, Supplementary Movie 5, right). Consistent with this, significantly more photoactivated flotillin-2 was observed in Rab5 and Rab11a compartments compared to Rab4 rapidly after photoactivation (Fig. 4d, magenta and green lines, 31.3 and 16.3 s, respectively). In line with our data showing

that flotillins play a greater role in sorting TCRζ into Rab11a compartments (Fig. 1), we observed a higher accumulation of internalised flotillin-2 in Rab11a—compared to Rab5-positive endosomes at later time points (Fig. 4d, black dashed line, 71.3 s). These results are consistent with the hypothesis that flotillin membrane domains constitute a link from the plasma membrane through an endocytic axis including Rab5-positive and Rab11a-positive endosomes.

**Flotillins regulate Rab5 and Rab11 endosome organisation**. Flotillin-rich membrane domains cluster and activate the molecular motor dynein to mediate transport along microtubules[39]. Dynein itself contributes to endocytic sorting by mediating the movement of endosomes towards the centre of the cell[21,22]. Thus, we hypothesised that if flotillins do indeed establish a fast lane for TCRζ sorting, these domains could also—via a potential connection to dynein—contribute to regulate the movement and distribution of sorting/recycling compartments in the cell. Consistent with this idea, we had previously observed that TCRζ-positive endosomes failed to be transported towards the cell centre in FlotKO T cells[14].

Here we investigated the spatial organisation of endogenous Rab4, 5 and 11a within WT and FlotKO Jurkat T cells. Quantification of Rab vesicle distribution in WT and FlotKO cells revealed that in absence of flotillins, Rab5 and Rab11a, but not Rab4, were significantly more dispersed throughout the cell and did not show the accumulation at the cell centre observed in WT cells (Fig. 4e, f). Flotillins therefore regulate the spatial organisation of endosomal compartments in a manner that would be consistent with a connection to dynein[22].

We next directly visualised TCRζ-GFP in transmission electron microscopy using APEX2[45] coupled to a GFP-binding peptide[46] to determine the contribution of flotillins to TCRζ-positive endosome structure. In WT Jurkat T cells, TCRζ endosomes were relatively uniform in size and shape (Fig. 4g, upper panel). By contrast, in FlotKO Jurkat T cells, TCRζ endosomes were very highly variable in size and shape, ranging from regular circular endosomes similar to WT, to tubular structures and large amorphous structures (Fig. 4g, lower panel). We observed a significant difference in variability of TCRζ-GFP vesicle circularity between WT and FlotKO cells (Fig. 4h), confirming that flotillins ensure a consistent architecture of TCRζ endosomes. Finally, we investigated if flotillins specifically regulated TCRζ endosome morphology or contributed to structural maintenance of the plasma membrane. We observed no difference in the organisation or integrity of the plasma membrane between WT and FlotKO cells (Supplementary Fig. 3a, b). Here again, the disruption of the architecture of TCRζ endosomes observed in absence of flotillins could result from disturbed membrane organisation and/or impaired positioning of these endosomes along microtubules. While these results do not prove a direct connection between flotillin-positive endosomes and microtubule motors, they do reveal a distribution of Rab5-positive, Rab11a-positive endosomes in FlotKO cells that is consistent with dynein-mediated spatial organisation of these endosomes.

Together, these results suggest that flotillins potentially regulate the sorting of TCR into a fast Rab11a recycling pathway through two distinct but potentially related mechanisms: firstly, by acting as entry doors along a Rab5 and Rab11a endocytic recycling pathway; secondly, by mediating the correct positioning of these compartments to function together as an endocytic recycling axis.

**Aggregation of Rab5 and Rab11 endosomes blocks TCR recycling**. Our data establish that flotillins regulate TCRζ sorting throughout the Rab5 to Rab11a endocytic recycling pathway,

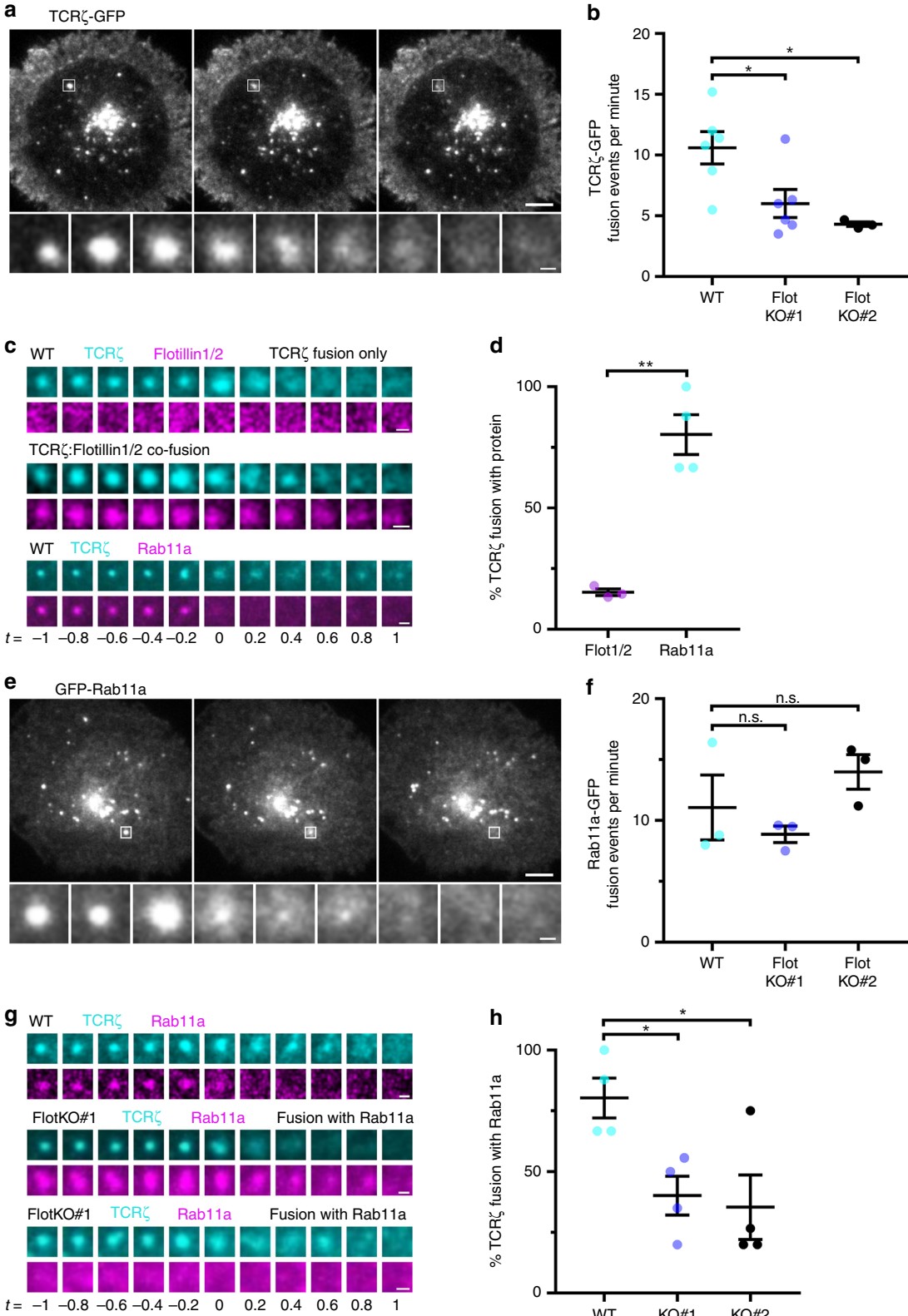

potentially via regulation of the spatial organisation of Rab5-positive and Rab11a-positive endosomes. We therefore sought to confirm the critical contributions of the spatial organisation of Rab5 and Rab11a to TCRζ recycling. Recent optogenetic aggregation methods effectively disrupt the subcellular distribution of Rab-GTPases using blue light[47]. We combined optogenetic aggregation of Rab5 and Rab11a with two-photon photoactivation

within endosomes to investigate if the spatial organisation of Rab11a endosomes supported the sorting of TCRζ for a return to the plasma membrane (Fig. 5a). Upon exposure to blue light, Cry2clust oligomerises, binding CIB1-tagged proteins to form large multimers[48]. Blue light-induced clustering of CIB1-Rab11a resulted in significantly less (Supplementary Fig. 4a, b) and more intense (Supplementary Fig. 4c, d) CIB1-Rab11a vesicles

**Fig. 3** Flotillins regulate TCRζ sorting into Rab11a vesicles for fusion with the plasma membrane. **a** Representative images of a TCRζ-GFP vesicle fusion event. Zoomed images below represent the indicated fusion event (white box) at 200 ms intervals. Main image scale bar = 5 μm, inset scale bar = 1 μm. **b** TCRζ-GFP vesicle fusion events per minute in WT and two FlotKO Jurkat T cell lines. **c** Representative images of TCRζ-GFP (cyan) vesicle fusion events with either flotillin-1 and flotillin-2 mCherry (magenta, top two panels) or mCherry-Rab11a (magenta, bottom panel). Scale bar = 0.5 μm. **d** Percentage of TCRζ-GFP fusion events containing either flotillin1/2-mCherry or mCherry-Rab11a. **e** Representative images displaying a GFP-Rab11a vesicle fusion event. Zoomed images below represent the indicated fusion event (white box) at 200 ms intervals. Main image scale bar = 5 μm, inset scale bar = 1 μm. **f** GFP-Rab11a vesicle fusion events per minute in WT and two FlotKO Jurkat T cell lines. **g** Representative images of TCRζ-GFP (cyan) vesicle fusion events with mCherry-Rab11a (magenta) in WT Jurkat T cells (top) or fusion with (middle) or without (bottom) mCherry-Rab11a in FlotKO Jurkat T cells. Scale bar = 0.5 μm. **h** Percentage of TCRζ-GFP fusion events containing mCherry-Rab11a in WT and two FlotKO Jurkat T cell lines. All images captured on a Zeiss ELYRA TIRF microscope. Data points indicate means of independent experiments, error bars indicate mean ± SEM. n.s. = not significant, *$p < 0.05$, **$p < 0.01$ from a two-tailed unpaired Student's T-test of means for TCRζ fusion of $n = 6$ (WT: 7; 6; 5; 10; 4; 5 cells, FlotKO#1: 3; 6; 3; 4; 6; 4 cells per experiment), 3 (FlotKO#2: 4; 5; 4 cells per experiment), $n = 3$ (Rab11a fusion, WT: 5; 5; 4 cells, KO#1: 5; 4; 3 cells, KO#2: 5; 5; 5 cells per experiment), 3 (flotillin1/2: 6; 5; 7 cells per experiment), 4 (Rab11a: 5; 5; 4; 5 cells per experiment), and 4 (TCR fusion with Rab11a, WT: 5; 5; 4; 5 cells, KO#1: 5; 5; 3; 5 cells, KO#2: 2; 3; 5; 4 cells per experiment) biologically independent experiments

compared to the WT-Rab11a control (Supplementary Movie 6), indicative of disruption in Rab11a spatial organisation.

Optogenetic aggregation caused no pronounced difference in the number of photoactivated TCRζ-PSCFP2 vesicles leaving the photoactivation region between CIB1-Rab11a and mCherry-Rab11a expressing cells (Fig. 5e). This indicates that optogenetic aggregation of Rab11a did not perturb TCRζ vesicle trafficking, but only disrupted the spatial organisation of Rab11a-positive endosomes. However, optogenetic aggregation of Rab11a-positive endosomes prior to two-photon photoactivation (Fig. 5a, b, Supplementary Movie 6) significantly reduced TCRζ recycling (Fig. 5c, d, Supplementary Movie 7). As with Rab11a, optogenetic aggregation of Rab5-positive endosomes (Supplementary Movie 6) did not disrupt TCRζ vesicular trafficking (Fig. 5i) but resulted in a significant reduction in TCRζ recycling (Fig. 5g, h, Supplementary Movie 7). Two-photon photoactivation of intracellular compartments combined with optogenetic aggregation revealed that the spatial organisation of Rab5 and Rab11a endosomes is critically important to TCRζ recycling. This highlights how maintaining the spatial organisation of an endocytic recycling axis, as is the case with flotillins, could ultimately contribute to efficient TCRζ recycling.

## Discussion

T cell activation relies on the recycling of internalised T cell receptor (TCR) to the plasma membrane at the immunological synapse[9,13,14]. The mechanisms that mediate the sorting of TCR into recycling endosomes are poorly understood. In this study, we have established a suite of live cell imaging and quantification tools based on single-photon and two-photon photoactivation of fluorescent proteins to gain insight into these sorting mechanisms. These tools allowed us to visualise and quantify every step of the endocytic recycling process, from endocytosis and sorting to delivery and fusion to the plasma membrane. Our data showed that TCR is recycled via a Rab5-Rab11a fast endocytic recycling axis. We further established that the membrane-organising proteins flotillins play a role in mediating TCRζ entry into and exit from Rab5 sorting endosomal compartments, ultimately regulating TCRζ transfer from Rab5 into Rab11a-positive endosomes. Indeed, without flotillins, TCRζ incorporation into Rab11a endosomes was greatly diminished, which resulted in reduced TCRζ vesicle fusion and delivery to the plasma membrane. We further evaluated the time taken by TCRζ to be returned to the plasma membrane from the moment it was endocytosed. Our findings show TCRζ recycling is consistent with the fastest observed recycling rates ($t\frac{1}{2} \sim 8.5$ min)[41], indicating flotillins support TCRζ sorting for a rapid Rab11a-mediated recycling. Furthermore, we found that flotillins are enriched along the Rab5–Rab11a endocytic axis, and they are required for the proper spatial organisation of these endosomal populations within the cell. Based on these findings, we propose that flotillins regulate the sorting of TCR into a fast Rab11a recycling pathway through two distinct but related mechanisms: by acting as membrane domain entry doors through the endocytic recycling network and by mediating the correct positioning of this network to control endocytic recycling.

Our data indicate that flotillins act at two stages in the TCRζ sorting pathway—first, entry into Rab5 endosomes and then transfer from Rab5 to Rab11a endosomes (Fig. 1). Surface proteins internalised into Rab5-positive endosomes can be either recycled, destined for degradation or for retrograde transport to the Golgi[1,18,49]. However, accumulating evidence suggests that cargoes committed to recycling can be segregated from late endosomal/retrograde-targeted cargo almost immediately following endocytosis, even before entry into Rab5-positive early endosomes[24,25,50]. For instance, transferrin (a Rab11-mediated recycling cargo) and epidermal growth factor (EGF, a cargo trafficked through a late endosomal pathway), are sorted into two distinct populations of early endosomes immediately after endocytosis[25]. We have found that flotillins mediate the entry of TCRζ, a clathrin independent cargo[6], into Rab5-positive endosomes within seconds following endocytosis, implicating them as potential regulators of pre-early endosomal sorting.

Our data shows that flotillins play a key role in sorting to the endocytic recycling pathway by mediating the transfer of TCRζ between Rab5 and Rab11a-postive endosomes (Fig. 1). Sorting between Rab5 and Rab11a compartments can occur via membrane tubulation[18], cargo transfer via membrane domains or a combination of both[51]. The microtubule motor dynein mediates sorting by facilitating transport of cargo towards recycling endosomes[21,23], interacting with membrane domain proteins (SNX4) and maintaining endosome spatial organisation[22]. Flotillins have been previously implicated in sorting for lysosomal-targeted cargoes in HeLa cells[52] and in regulating Rab11a-dependent recycling[34–36]. Importantly, flotillins are well established membrane domain proteins[28,31,32,40] that interact with SNX4[35]. Furthermore, flotillin-rich membrane domains cluster and activate the microtubule motor dynein[39]. Flotillins therefore interact with a subset of protein and membrane domains that have been established to mediate sorting along the endocytic recycling pathway.

Altogether, our data are consistent with a model in which flotillins provide areas of cellular membranes with a distinct identity (Fig. 6). Hence, we propose that this distinct flotillin-mediated membrane identity is carried throughout endocytic recycling and establishes a fast lane for the return of proteins internalised in flotillin-rich vesicles to the cell surface. In this model, flotillins either interact directly with machinery required

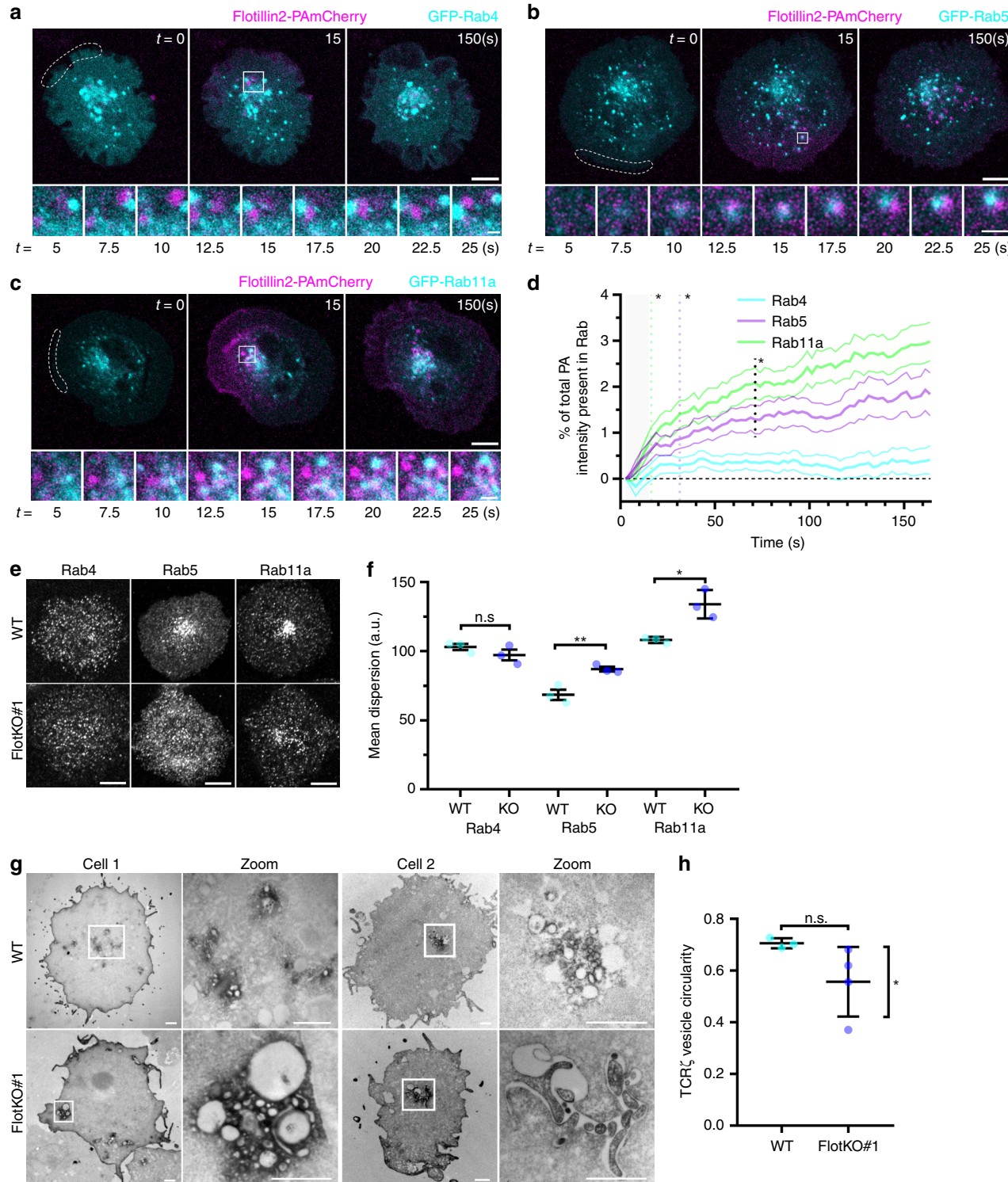

for endosomal sorting and transport (as supported by refs. [37,38]) or regulate the membrane domain composition to facilitate recruitment of these factors (as supported by ref. [39]).

When proteins localised in flotillin-rich membranes reach Rab5-positive endosomes, they are prioritised for rapid sorting into Rab11-positive endosomes. In our proposed model, only a small minority of Rab5 endosomes contains flotillin domains, but a majority of these flotillin-positive Rab5 membranes become Rab11a-positive endosomes. The physical power for this sorting mechanism along the Rab5–Rab11a axis is provided by the ability of flotillin-rich domains to cluster and activate the microtubule

motor dynein[39]. Hence, flotillin-rich domains would simultaneously define the identity of this endocytic axis and couple it to the motor that supports its operation. While this model is consistent with all of our observations and with the literature, we have not formally identified flotillin-rich membrane domain at the surface of Rab5 and Rab11-positive endosomes or shown a direct link between flotillins and dynein. Super-resolution techniques such as 3D single molecule localisation microscopy and correlative light and focussed ion beam scanning electron microscopy would help to delineate flotillin-scaffolded membrane domains on Rab5 and Rab11-positive endosomes. Additionally,

**Fig. 4** Flotillins rapidly interact with and maintain the organisation of Rab5 and Rab11a-positive endosomes. **a** Representative images of flotillin-2-PAmCherry photoactivated at the dashed region failing to interact with EGFP-Rab4 endosomes in WT Jurkat T cells. Cells were photoactivated for five frames with 2.5 s intervals, then imaged every 2.5 s for 60 frames. **b** Representative images of flotillin-2-PAmCherry photoactivated at the dashed region rapidly interacting with EGFP-Rab5 endosomes in WT Jurkat T cells. Cells were photoactivated and imaged as in **a**. **c** Representative images of flotillin-2-PAmCherry photoactivated at the dashed region rapidly interacting with EGFP-Rab11 endosomes in WT Jurkat T cells. Cells were photoactivated and imaged as in **a**. For **a**–**c** zoomed images below are indicated by a white box, main image scale bar = 5 µm, inset scale bar = 1 µm, and flotillin-2-PAmCherry was co-expressed with untagged flotillin-1. All cells imaged on a Zeiss 880 confocal, photoactivation performed at 405 nm. **d** Quantification of the percentage of photoactivated flotillin-2-PAmCherry present in Rab4, 5 or 11a. **e** Representative confocal images of endogenous Rab4, Rab5 and Rab11a detected with corresponding antibodies in WT (top) or FlotKO (bottom) Jurkat T cells. Scale bar = 5 µm. **f** Mean fluorescence dispersion of the indicated markers. Data points indicate mean dispersion from $n = 3$ biologically independent experiments (Fig. 4a–d, Rab4: 10; 8; 5 cells, Rab5: 10; 12; 10 cells, Rab11a: 9; 11; 10 cells per experiment) (Fig. 4e, f, Rab4 WT: 10; 10; 10 cells, Rab4 KO#1: 9; 11; 10 cells, Rab5 WT: 11; 10; 10 cells, Rab5 KO#1: 10; 10; 10 cells, Rab11a WT: 10; 10; 11 cells, Rab11a KO#1: 10; 10; 10 cells per experiment). **g** Representative transmission electron microscopy images of DAB-reacted APEX2-GBP co-expressed with TCRζ-GFP in WT (top panel) or FlotKO (bottom panel) Jurkat T cells. Scale bar = 2 µm. **h** APEX-GBP-labelled vesicle circularity. Data points indicate mean circularity from $n = 3$ (WT: 4; 3; 1 cells per experiment) and 4 (FlotKO#1: 7; 1; 2; 2 cells per experiment) biologically independent experiments of 1–4 cells. $*p < 0.05$ from $F$-test to compare variance. Error bars indicate mean ± SEM. n.s. = not significant. $*p < 0.05$ from Wilcoxon rank sum test **d**; n.s. = not significant, $*p < 0.05$, $**p < 0.01$ from a two-tailed unpaired Student's $T$-test of independent experiment means **f**. n.s. = not significant from a two-tailed unpaired Student's $T$-test of independent experiment means and $*p < 0.05$ from $F$-test of variance **h**

proximity labelling of the flotillin proteome would confirm interactions between flotillins and dynein required for the sorting step that we revealed in this study. Such experiments would answer the questions that remain to be solved in order to fully demonstrate the model that we propose here: what is the mechanism allowing TCRζ to be internalised together with flotillin; are there really flotillin-rich domains on the surface of endosomes in T cells; do all flotillin-positive Rab5 membranes become Rab11a-positive; and do flotillin domains really interact with dynein during recycling, and if yes, how?

## Methods

**Plasmids**. Expression constructs encoding for human flotillin-1 or flotillin-2 were a gift from V. Nigli (Univerity of Bern). TCRζ-PSCFP2 was provided by Prof. K. Gaus (University of New South Wales). PA-mCherry expression backbone was obtained from Clontech. TCRζ PA-mCherry and flotillin2-PAmCherry were generated by inserting TCRζ/flotillin2 into the PAmCherry backbone with EcoRI and AgeI[14]. mCherry-Rab5 was a gift from Gia Voeltz (Addgene plasmid # 49201). mCherry-Rab7a was a gift from Michael Davidson (Addgene plasmid # 55127). mCherry-Rab11a was a gift from Michael Davidson (Addgene plasmid # 55124). CIB1-FuGeneRed-Rab11a and CIB1-FuGeneRed-Rab5 were gifts from Prof. Won Do Heo (Korea Advanced Institute of Science and Technology). mCerulean-Cry2clust was designed based on the sequences provided in ref. [48] and commercially sourced from Integrated DNA Technologies.

**Antibodies**. Anti-Rab4 (1:100, Cell Signaling Technology Cat# 2167P, RRID: AB_10827896), anti-Rab5 (1:200, Cell Signaling Technology Cat# 3547, RRID: AB_2300649 CST#3547), anti-Rab11a (1:50, Cell Signaling Technology Cat# 2413S, RRID:AB_2173452), and anti-rabbit Alexa647 (Thermo Fisher Scientific Cat# A-21245, RRID:AB_2535813).

**Cell culture**. Jurkat T cells (ATCC Cat# TIB-152, RRID:CVCL_0367) and flotillin-1 and flotillin-2 knock-out Jurkat cell lines were cultured in RPMI 1640 medium (Gibco) supplemented with 10% (vol/vol) FBS, 2 mM L-glutamine and PenStrep (all from Invitrogen). Flotillin-1 and flotillin-2 knock-out Jurkat cell lines were generated using two guide RNAs each targeting the genomic DNA of flotillins 1 and 2 together with Cas9 expression plasmid. Transfected single cells were FACS sorted and seeded into 96-well plates and screened for flotillin1/2 knockout by western blot[14]. Cells were transfected with 1 µg DNA per 200,000 cells, 18–20 h prior to imaging using the Neon electroporation kit (Invitrogen).

Before imaging, cells were incubated for 10 min at 37 °C on 18 mm glass-coated surfaces (Marienfeld) that were prepared by incubating with poly-L-lysine (Sigma) for 30 min at room temperature, then 1 µM anti-CD3ε (16-0037; eBioscience) and anti-CD28 (16-0289; eBioscience) antibodies for T cell activation. For live cell imaging, cells were imaged from 10 to 40 min after their deposition on the coverslips.

For immunostaining, cells were fixed with 3.7% EM-grade paraformaldehyde (C004, ProScitech) for 30 min at 37 °C. After fixation, cells were permeabilized with 0.15% triton-X100 (Sigma), blocked in 5% BSA and probed with primary and secondary antibodies sequentially.

**Microscopy**. Fixed and live-cell confocal microscopy were performed on a Leica SP5 (Leica, Germany; Leica LAS AF Image Acquisition Software, RRID:

SCR_013673) or Zeiss LSM780 laser-scanning confocal microscope (Zeiss, Germany; ZEN Digital Imaging for Light Microscopy, RRID:SCR_013672) that are equipped with an argon laser (405, 488 nm), a diode pump solid state laser (561, 647 nm), and a live-cell incubation chamber (Pecon). Two-photon photoactivation microscopy was performed on a Zeiss LSM 880 laser-scanning confocal microscope (Zeiss) equipped with a Mai Tai Insight DeepSee tuneable multi-photon laser in addition to the above mentioned.

Live-cell TIRF images were acquired on a total internal reflection fluorescence microscope (ELYRA; Zeiss) with a ×100 oil-immersion objective with a numerical aperture of 1.46. Images were captured with a 20 ms exposure time. To visualise vesicle fusion events, plasma membrane signal was first bleached for 2 s with maximum 488 nm laser power prior to imaging.

GFP constructs and photoactivated TCRζ-PSCFP2 were excited using the 488 nm line of the argon laser source, while PA-mCherry, mCherry and FuGeneRed tagged proteins were excited with the 561 nm laser line. Images were acquired with a ×100 1.4NA DIC M27 Apo-Plan oil immersion objective (Zeiss, Germany) and GaAsP-PMTs in simultaneous, bidirectional scanning mode. For each channel the pinhole was set to 1 Airy Unit.

Confocal photoactivation was achieved by illuminating a region of the cell outer membrane with a 7.2 µW 405 nm laser pulse with 12.24 µs pixel dwell time.

Two-photon photoactivation was achieved by illuminating pericentriolar-localised TCRζ-PSCFP2 at a focal point 2.5 µm from the coverslip with a 58 mW 800 nm laser pulse with 65.54 µs pixel dwell time every 30 s for 10 consecutive cycles.

Optogenetic aggregation was achieved by exposing the entire cell to 488 nm laser light with a pixel dwell of 8.19 µs for three frames, with a 1 µm Z-gap between frames, over a 30 s period for 10 cycles.

**Electron microscopy**. Cells co-transfected with TCRζ-GFP and APEX2-GBP were activated for 20 min on 35-mm gridded dishes (MatTek Corporation), followed by immediate fixation with 2.5% glutaraldehyde in 0.1 M sodium cacodylate for 1 h at room temperature. Cells were then washed, incubated with 3,3′-diaminobenzidine (DAB) (Sigma Aldrich) and incubated for 30 min at room temperature with a DAB reaction buffer containing $H_2O_2$. Post-fixation was carried out for 10 min with 1% $OsO_4$ (osmium tetroxide) (ProScitech). This was followed by serial dehydration in increasing percentages of ethanol. Cells were then serially infiltrated with LX112 resin in a BioWave microwave (Pelco). Fresh 100% resin was then added and polymerised at 60 °C overnight. Ultrathin sections were cut on an ultramicrotome (UC6: Leica) and imaged at 100 kV on a JEOL1400 transmission electron microscope.

**Image analysis**. Vesicle count and cross-channel nearest-neighbour distances for TIRF and two-photon experiments were determined with a custom Matlab (MATLAB, RRID:SCR_001622) vesicle tracking and cross-channel nearest-neighbour distance evaluation software. For vesicle counts and cross-channel nearest-neighbour distance determination following two-photon photoactivation, the mCherry signal was used as a mask to exclude the pericentriolar photoactivated region. A GUI application and all source code for this analysis is freely available from at: https://github.com/PRNicovich/PAVesT.git.

Circularity measures were performed using FIJI (Fiji, RRID:SCR_002285)[53]. A threshold was set to identify the lumen of APEX2 reaction product defined endosomes, following which circularity was determined using the analyse particles tool.

Vesicle fusion events per minute were quantified from singly transfected cells by visually identifying fusion events in a blinded manner. Co-fusion events were visually identified, and counted as containing the indicated protein if a decrease in

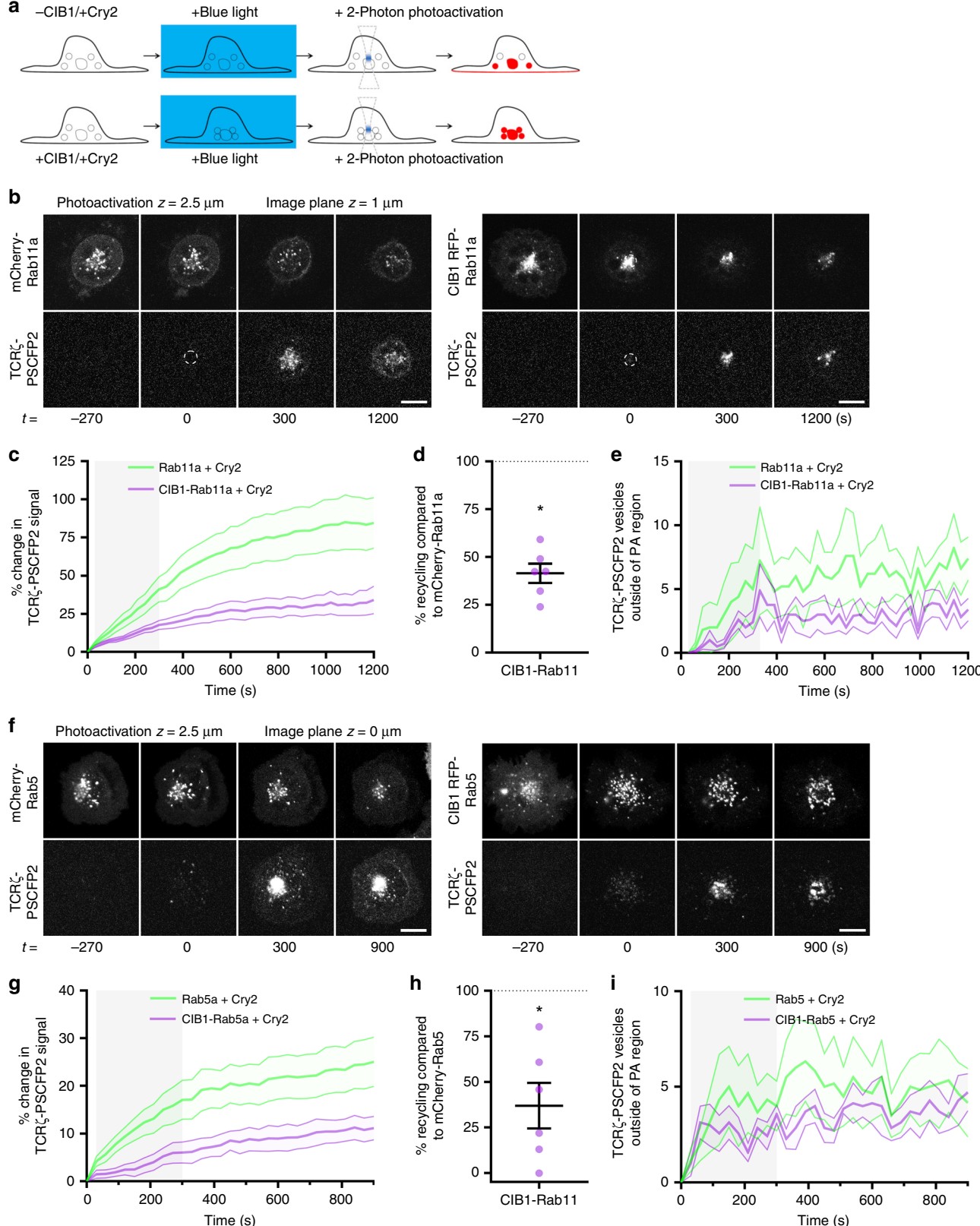

mean maximum intensity from the last three frames was >15% (TCRζ) compared to the maximum intensity of the initial frame as quantified by the analyse particles measure in FIJI.

Mean dispersion was calculated using a custom FIJI plugin, which evaluates endosome dispersion by calculating the intensity weighted measure of the average pixel distance from the centre of mass of the cell. The plugin is available on request.

Recycling analysis was performed using FIJI. A photoactivated endosome mask was created via thresholding, subtracted from the PSCFP2 channel to ensure only plasma membrane signal was present, and plasma membrane PSCFP2 signal quantified.

TCRζ entry into Rab5/11 endosomal compartments was quantified in FIJI by creating a thresholded mask of endosomal compartments using the Rab and PA-mCherry channel, the background set to zero, then the PA-mCherry channel

**Fig. 5** Rab5 and Rab11a are critical to TCRζ recycling. **a** Schematic depicting the coupling of optogenetic aggregation using CIB1/Cry2 with two-photon photoactivation. **b** Representative images of TCRζ-PSCFP2 photoactivated from either mCherry-Rab11a (left) or CIB1-FugeneRed-Rab11a (right) co-expressed with mCerulean-Cry2clust following exposure to blue light for 10 frames with 30 s intervals. Two-photon photoactivation was undertaken 2.5 µm within the cell for 10 frames, photoactivation region indicated by dashed circle. **c** Photoactivated TCRζ-PSCFP2 signal over time at the plasma membrane when photoactivated from CIB1-FugeneRed-Rab11a or mCherry-Rab11a compartments when co-expressed with Cry2clust and exposed to blue light. **d** Mean TCRζ-PSCFP2 signal from 900 to 1200 s timepoints when photoactivated from Cry2clust-aggregated CIB1-FugeneRed-Rab11a, normalised to mCherry-Rab11a as 100%. **e** Photoactivated TCRζ-PSCFP2 vesicle number outside the initial photoactivated region over time when photoactivated from CIB1-FugeneRed-Rab11a or mCherry-Rab11a when co-expressed with Cry2clust and exposed to blue light. Image plane quantified is 1 µm above the plasma membrane. **f** Representative images of TCRζ-PSCFP2 photoactivated from either mCherry-Rab5 (left) or CIB1-FugeneRed-Rab5 (right) co-expressed with mCerulean-Cry2clust following exposure to blue light for 10 frames with 30 s intervals. Two-photon photoactivation was undertaken 2.5 µm within the cell for 10 frames, photoactivation region indicated by dashed circle. **g** Photoactivated TCRζ-PSCFP2 signal over time at the plasma membrane when photoactivated from CIB1-FugeneRed-Rab5 or mCherry-Rab5 compartments when co-expressed with Cry2clust and exposed to blue light. **d** Mean TCRζ-PSCFP2 signal from 900 to 1200 s timepoints when photoactivated from Cry2clust-aggregated CIB1-FugeneRed-Rab5, normalised to mCherry-Rab5 as 100%. **e** Photoactivated TCRζ-PSCFP2 vesicle number outside the initial photoactivated region over time when photoactivated from CIB1-FugeneRed-Rab5 or mCherry-Rab5 when co-expressed with Cry2clust and exposed to blue light. Image plane quantified is 1 µm above the plasma membrane. Data points indicate means of independent experiments, error bars indicate mean ± SEM. *$p < 0.05$ from one sample $t$-test comparing values to normalised mCherry-Rab5/Rab11a mean of 100% from $n = 6$ biologically independent experiments (mCherry-Rab11a: 1; 1; 1; 1; 1; 1 cells, CIB1-FugeneRed-Rab11a: 1; 1; 1; 2; 1; 1 cells, mCherry-Rab5: 2; 2; 1; 2; 2; 1 cells, CIB1-FugeneRed-Rab5: 3; 1; 1; 3; 1; 1 cells per experiment). Scale bar = 5 µm

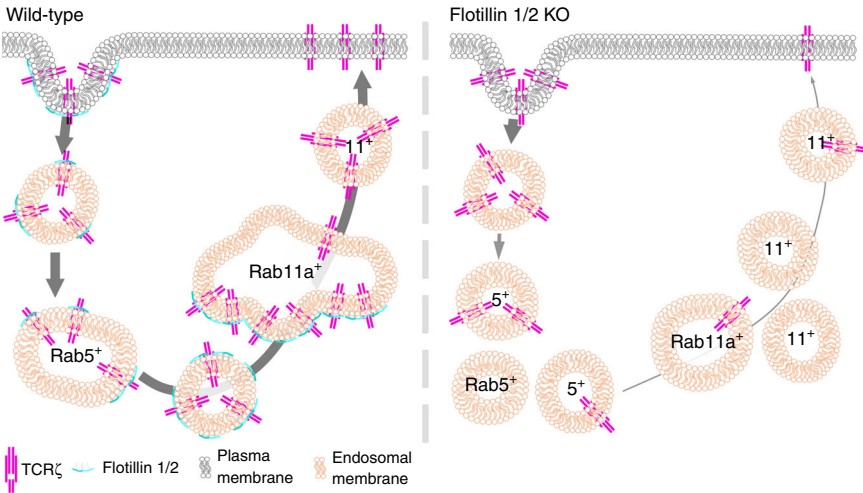

**Fig. 6** Schematic of flotillin regulation of receptor sorting and recycling. Following endocytosis in WT cells (left), TCRζ is efficiently sorted from flotillin-positive endosomes into Rab5-positive endosomes via rapid interaction, then transferred into Rab11a-positive endosomes via flotillin-positive endosomes and recycled to the plasma membrane within Rab11a-positive vesicles in an flotillin-independent manner. In flotillin 1 and 2 knockout cells (right) TCRζ endocytosis is unaffected, but TCRζ entry into, and the spatial organisation of, Rab5-positive and Rab11a-positive endosomes is perturbed, leading to reduced TCRζ entry into Rab5-positive endosomes and large reduction of TCRζ transfer from Rab5 into Rab11a-positive endosome, resulting in impaired TCRζ recycling to the plasma membrane

divided first by the PAmCherry mask, then the Rab mask to calculate the total endosomal PA-mCherry intensity present in Rab compartments.

**Statistical analysis**. All statistical analysis excepting time of divergence analysis were performed using GraphPad software (Graphpad Prism, RRID:SCR_002798). Statistical significance between datasets was determined by performing two-tailed, unpaired non-parametric Students $T$-tests or one sample $T$-tests, where stipulated. Graphs show mean values, and error bars represent the SEM. In statistical analysis, $p > 0.05$ is indicated as not significant (n.s.), whereas statistically significant values are indicated by asterisks as follows: *$p ≤ 0.05$, **$p < 0.01$, ***$p < 0.001$.

Time of divergence analysis was performed using a custom MatLab script. To determine the time-point of divergence, pooled fluorescence intensity distributions of WT and FlotKO#1 were compared by means of the non-parametric Wilcoxon rank-sum test at each time point. Briefly, the null hypothesis that WT and FlotKO#1 data sets are from continuous distributions with equal medians is tested against the alternative hypothesis they are not. For each time-point a $p$-value is calculated, with the time point of divergence defined as the first time-point after which all subsequent $p$-values are equal to or smaller than the considered significance level. *$p ≤ 0.05$, **$p < 0.01$.

## Data availability
All confocal and electron microscopy data (Figs. 1, 2, 4 and 5) is deposited on FigShare and is publicly available: [https://figshare.com/projects/_Flotillins_promote_T_cell_receptor_

sorting_through_a_fast_Rab5-Rab11_endocytic_recycling_axis_raw_microscopy_data/66398]. TIRF data (Fig. 3) and all data that support the findings of this study are available from the corresponding author upon reasonable request.

## Code availability
MATLAB codes are available from the author's GitHub repo [https://github.com/PRNicovich/PAVesT.git]. FIJI plugins are also available upon request.

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

## Acknowledgements

We would like to thank the staff of the BioMedical Imaging Facility of the University of New South Wales, the facilities supported by AMMRF at the Electron Microscopy Unit at UNSW and Dr. Maite Vidal-Quadras for insightful discussions. We thank the funding bodies: National Health, Medical Research Council (APP1102730), Australian Research Council (DE140101626), and Swiss National Science Foundation (31003A_172969).

## Author contributions

G.M.I.R., M.E., N.K.-K., H.V. and N.A. performed the experiments. G.M.I.R., M.E. and D.K. analysed the data. M.C. and D.K. established the analysis methods. M.B. and G.M.I.R. contributed to data interpretation and manuscript writing. J.R. designed the project and wrote the manuscript.

## Competing interests

The authors declare no competing interests.
