## [peer review file · Nature Communications]

Reviewers' comments:

Reviewer #1 (Remarks to the Author):

This is generally a nice paper.

My only criticism of substance is that too much of the analysis of flotillin KO Jurkat cells is apparently done on one clone of the cells. There is a considerable risk of clone-dependent artifact. This is especially the case as there is no plausible mechanism in the paper by which flotillins may have their effect. It seems quite likely that different clones of Jurkat cells might have different endosomal morphology or recycling rates, given the selective pressures inherent in making KOS.

I strongly suggest that key experiments - especially the morphological examination of endosomes in Figure 1, and data on endosomal recycling in Figure 7 are both done in multiple clones and rescued by re-expression of flotillins.

Reviewer #2 (Remarks to the Author):

Review:

Redpath et. al. presented the important role of flotillins that regulate spatial organization of specific endosomal recyclings, specifically TCR sorting into Rab 11. The authors demonstrated their work by comparing with wild-type and flotillins knock-out cell lines and also developed a new method to visualize components in the recycling endosomes to the plasma membrane in real-time. Using the developed system, the authors dissected the regulation and importance of flotillin in Rab5 and Rab 11 mediated endosomal activities. Their novel system seems to be useful in the future but their findings are hard to understand and agree with from the current results. The developed model needs further characterization and their results contain a number of critical problems need to be accomplished. Also, they need serious modification of their paper.

Major:

Fig. 1: 1b, 1d: The authors demonstrate that without flotillins, TCR positive endosome around MTOC is impaired. The data demonstrate that accumulations at specific regions are impaired but does not demonstrate where the MTOC is exactly. To strengthen their point, they should observe with MTOC marker, e.g. using the centrosome marker.

Fig. 2c: It is hard to observe the differences through the image. The color should be changed to black and white or inverted. Also, as demonstrated in 2c, it seems like mCherry tagged Flotillin1/2 was overexpressed in WT. I wonder why they did not use the KO cell line to minimize the secondary effect as possible. Numerous reports demonstrate the problems with overexpressing Flotillin. To further support their finding, they should rescue the Flotillin in KO cell to observe the fusion of TCR positive endosome.

Fig. 2d: The result is confusing. On the previous report, the author's group demonstrated the importance of flotillins in TCR recycling. However, this result seems like in TCR recycling the independent of flotillin existence. The authors should clarify their result in more detail.

Fig. 3a, 3b: From Fig. 2d, the result demonstrated the TCR recycling is more related to Rab11a existence rather than with Flotillins 1/2. If 3b demonstrated that Flot KO cell reduced the fusion of Rab11-GFP, then this makes more sense that TCR fusion event was disrupted without Flot. In fact, there was no problem with Rab11 fusion event with KO cell. The authors should elaborate more in detail on this matter.

Fig. 3: In this figure, the authors concluded the results that without flotillins, TCR is poorly sorted into Rab11 cycling vesicles. The authors should strengthen their point by demonstrating the fusion event with different controls. In Fig. 1, the localization of LACT-C2 (phosphatidylserine marker), Rab5 and Rab11 was affected but not with phosphoinositide PIP3, PIP4, Rab4 and Rab8. If their finding is specific for Rab11, than fusion events, especially with Rab4, should not be affected. Also try with Rab 11 mutants.

Since one of the strong points of this paper is the development of a novel method to visualize endosome fusion event, they should characterize this system in more detail by demonstrating how spatially the photo-activation can be reached to, e.g. range of spatial activation. Also the how long does it take for the activation of the molecule to be visualized.

Fig. 4c-f: If the result is really specific to the existence of Flotillins, then other than Rab7 additional experiments with other controls are necessary (those that was not affected by Flot KO in Fig. 1). If Rab11 is the important factor, then the mutation of Rab11 should be added. Also why did Rab11a demonstrate higher TCR recycling event, while LACT-C2 and TCR increased the pixel in a similar manner? Rather than Student's t-test, the ANOVA with Tukey multiple comparison tests is more suitable to compare between different conditions.

Fig. 4g: "...TCR recycles from a flotillin-positive, Rab11a-positive..... but not flotillins" This sentence is hard to understand. What are the authors trying to say? It is also hard to understand the data from the legend and the main text. I guess the authors are trying to say is that photo-activation where flot1/2 was accumulated demonstrated recycling of TCR. But, no synchronized signal with Flot1/2 was observed but synchronize signal with Rab11 or LACT-C2 with TCR containing vesicle was observed. Again, the different control group is necessary to strengthen their points.

☐ Fig. 5g: The photo-convergence fluorescent protein used in Fig. 4 and Fig. 5 are different. Therefore, it is not suitable to calculate how long TCR vesicle took for recycling, unless they quantify the photo-activated time of two fluorescent protein.

☐ Fig. 5h: The authors conclude that flotillin is critical to sort TCR into Rab5 to transfer into Rab11. Up to this point, the authors only focused on the activity of Rab 11. But this data demonstrate the importance of Rab 5 that consequently affected sorting into Rab 11. This data implies the previous results are not the important factor but consequent result from upstream events. Authors should focus more data with Rab5 in previous figures or else, the previous results seemed to be impractical.

☐ In the chapter where Fig. S2 is present. Although this part is divided into a subgroup, there is no new information but just validating what was presented in previous studies with their technique. Also the cell they used was not presented. They mentioned that TCR reached phosphatidylinositol-positive endosomes no longer than 7 seconds. Is this time mediated by Rab 5? Because on the previous chapter, they mentioned 6.5-7 minutes for TCR to return to PM. I am guessing 7 seconds is from when TCR is first internalized. Further information is needed.

☐ Fig. S3: In order to observe the spatial aggregation of Rab11, then the spatial light stimulation will be more adequate than whole cell stimulation.

☐ Fig. 6b: there seemed to be no cluster formation other than the aggregation pattern due to Rab11 expression. To strengthen their point, they demonstrate the spatial light stimulation to generate aggregation on a specific location then perform their two-photon visualization modules. Also, they should compare with KO cell.

☐ Fig. 6: The authors concluded that spatial organization is important for recycling endosome. But Fig. 5h, the authors demonstrated that without flotillin, the sorting into Rab5 was also problematic. Therefore, they should also demonstrate the importance of flotillin with Rab5-mediated activity.

☐ Fig. 7: It is hard to conclude that flotillin is universal regulators of endocytic cargo sorting since the authors are only looking at transferrin endocytosis activity. They should dissect other endocytosis activity. Also because Rab 5 is also perturbed, it is not reasonable to conclude that Rab 11 mediated activity is affected by flotillin. Rab11 sorting can be disrupted if Rab5 sorting has already defected.

Minor:

☐ Fig. 1: Figure numbering is poorly matched with the main text. As for Fig. 1, there are two b.

☐ Fig. 2a: Other than WT cell only, representative images or movie of reduced fusion events in the KO cells should be included, besides the quantified data.

- ☐ Fig. 3c, 3d: The fluorescence images are better than 2c but color changes might be better for better elucidation.
- ☐ Fig. 3f: There seem to be significant differences between two KO cell lines. The significance should be provided or the number of cells should be increased.
- ☐ Supplementary Fig. S1 is missing. S2 should be label into S1.
- ☐ Supplementary Fig. S2: Although will not easily happen with two-photon activation, the bleaching effect should be characterized, especially because the results are largely based on the fluorescence signal differences.
- ☐ Fig. 4b: It is understandable that the authors are targeting where TCR signals are accumulated. But if they are specifically targeting at the pericentriolar microtubule organization complex as they stated in their main text, they should demonstrate this region specific photo-activation, e.g. with marker staining. Also, the signal starts to appear the 30s after photo-activation, will this be changed with different activation condition?
- ☐ Fig. 5a: no description in the main text.
- ☐ Fig. 5b: The convergence of mCherry and GFP signal is not clear. The magnified inset is necessary.
- ☐ Fig. 5c and 5f: It is hard to see the divergence of the dashed line. The magnified graph is necessary.
- ☐ Fig.5: "Following two-photon photoactivation,..." This sentence will be better to be in Fig. 4e.

Reviewer #3 (Remarks to the Author):

This paper provides new information on TCR sorting and recycling, and importantly, reveals a role of the flotillin proteins in cargo recycling (that was proposed earlier by the present group of authors and in earlier publications by other groups).

The central and novel observation in this paper is a disruption of the spatial organization of recycling endosomes in Jurkat T cells when flotillins are knocked out (ko). The authors show a reduction in TCR vesicle fusion in Flot ko cell lines compared to wt from average 10 events (?) in wt to average 5 events (?) in Flot ko per minute. Flot ko reduced the amount of TCR in Rab11 positive vesicles fusing with the membrane by about 40%. The authors conclude: `This implies that the TCR is poorly sorted into Rab11a recycling vesicles and consequently fails to be delivered to plasma membrane.`

Given that there is a reduction but not failure of TCR delivery to the PM, the above statement is a bit too strong!

The authors use sophisticated methods which they established in their recent paper (Nat Comm, 2018) to visualize intracellular vesicle trafficking and cargo recycling. They succeed to activate specifically the fluorescence of TCR-PSCFP2 assembled near the centrosome, the position of the recycling compartment, thus avoiding confusion with signals from the PM.

Given that TCR is more widely dispersed in Flot ko cells, how can they be as precise in activating TCR at the centrosome in Flot ko cells?

Anyhow, with this method, they show a rapid recycling of the TCR back to the membrane (6-7min). In Flot ko cells, TCR was less efficiently transferred into Rab5 and Rab11 positive endosomes but was still occurring! This prompts the question what is the role of flotillins?

The authors control for the specificity of this defect in TCR recycling by following transferrin recycling, and show that the transferrin present in Rab5 endosomes is reduced in Flot ko cells (70 % compared to 44% reduction of TCR). Transfer to Rab11 vesicles per se is not impaired but fusion with PM (40 and 20% in Flot k.o. versus 75% in WT).

Thus, there is a significant role of Flot1 and 2 in TCR and Tf recycling which, in case of Tf, is consistent with earlier results by other groups (Solis et al., 2013).

The authors state that the spatial organization of the endosomal system is disrupted by Flot ko, and the images illustrating this effect are convincing. This is a nice result. However, it remains entirely unclear how flotillins would mechanistically contribute to establishment or maintenance of the spatial organization of the Rab5 and Rab11 positive endosomes. Flotillins are localized to the cytoplasmic face of the vesicle membranes. What is their idea how flotillins as microdomain forming membrane associated proteins regulate the spatial organization of the pre-early endosome and recycling compartment?

Flotillins are abundant at the Rab11 recycling compartment (Gagescu et al 2000; Solomon et al., 2002) and less so at Rab5 positive endosomes.

The authors show that flotillin vesicles are positive for phosphatidyl serine (PS) and Rab11a but are they Rab5 positive as is shown on their scheme in Fig.8? Should the arrow from the putative pre-early endosome not rather point to the transit from Rab5 to Rab11 (bypass Rab5)? This needs to be experimentally clarified. What exactly do they mean by pre-early endosome? This aspect (see also Discussion lines 357 to 363) remains vague and controversial (...sorting prior to entry into Rab5...).

That flotillins might mediate the interaction with cytoskeletal elements is discussed towards the end of the manuscript but not shown in the scheme (Fig.8). It seems to me that the reader needs more information as to how the spatial order of endosomes is perturbed in Flot knock out cells and how cargo sorting is affected.

The authors use Jurkat cells in which flotillin-1 and 2 were knocked out. They concentrate on the analysis of the TCR recycling pathway. They do not comment on which other aspects the Jurkat T cells may become abnormal (or remain normal ?) when flotillins are missing. From work by other

groups flotillins are known to form a preformed cap opposite from the MTOC/centrosome and recycling compartment. This is not included in their scheme of Fig 8, and is mentioned no-where in the text. This is to say that probably additional aspects of disorder might occur in the absence of flotillins. And the absence of flotillins from the PM might have severe effects.

The role of flotillins in sorting and recycling has been shown earlier by other investigators and needs to be cited appropriately. Their presence near the centrosome has also been demonstrated earlier as well as the interaction of flotillins with the cytoskeleton. Proper citations are necessary. That flotillins are also known under reggies and that knock out mice are existing needs to be mentioned as well. Is any sort of defect related to the present results observed in flotillin knock out mice?

Altogether, the manuscript contains highly interesting results concerning TCR recycling and concerning the role of flotillins in intracellular trafficking. But it seems that more work is needed towards the mechanism of flotillin functions.

Rebuttal letter for “Flotillins promote T cell receptor sorting through a fast Rab5-Rab11 endocytic recycling axis” (NCOMMS-18-23454-T)

Point-by-point response to referees’ comments

Reviewer #1

This is generally a nice paper.

We first would like to highlight that in light of the reviewers’ comments and new data, we have substantially reorganised and streamlined the structure of the manuscript. This revised version is more compact and more focused on illustrating the mechanism by which flotillins contribute to sort TCR for recycling. For these reasons, the new manuscript no longer includes the data on phosphatidylserine and on transferrin recycling, which remain valid and relevant and will be part of a separate manuscript. We believe these changes, which have been made with the reviewers’ comments in mind, truly make the manuscript better and clearer. The figure numbers mentioned in the response to the reviewers refer to the new version of the manuscript.

My only criticism of substance is that too much of the analysis of flotillin KO Jurkat cells is apparently done on one clone of the cells. There is a considerable risk of clone-dependent artifact. This is especially the case as there is no plausible mechanism in the paper by which flotillins may have their effect. It seems quite likely that different clones of Jurkat cells might have different endosomal morphology or recycling rates, given the selective pressures inherent in making KOS.

I strongly suggest that key experiments - especially the morphological examination of endosomes in Figure 1, and data on endosomal recycling in Figure 7 are both done in multiple clones and rescued by re-expression of flotillins.

We understand the reviewer’s concern and have therefore performed the experiments whose results were essential to the general message of the manuscript with a second clone. We included data collected on two flotillin knock out Jurkat clones for: **a)** experiments showing the importance of flotillins in sorting TCR after endocytosis into and out of Rab5 -positive endosomes (Fig. S1); **b)** experiments showing that without flotillins, the return from intracellular compartment to the plasma membrane of TCR was impaired (Fig. S2); and **c)** experiments showing that less Rab11a-positive vesicles fusing with the plasma membrane contained TCR in absence of flotillins (Fig. 3e-h). We obtained consistent data across both FlotKO clones in each of these experiments, which indicates that the function of flotillins in promoting the sorting of internalised TCR into a Rab5-Rab11a endocytic recycling axis is not a feature of one particular clone.

We further agree with the reviewer's comment regarding the fact the mechanism explaining the function of flotillins was unclear in the previous version of the manuscript. This is also the main reason why we have decided to reorganise the way and order the data are presented in the new version. We hope the mechanism describing flotillin function in recycling TCR is now more obvious throughout the manuscript. In order to better define this mechanism, we have also performed additional experiments to understand the potential role of flotillin in establishing a link from the plasma membrane to Rab11a-positive compartments (Fig. 4 a-d). Finally, we have included a proposed model at the end of the discussion, which comprehensively describes the mechanism through which we think flotillins contribute to sort internalised TCR into a Rab5-Rab11a recycling pathway (page 32, lines 389-404).

Reviewer #2

Redpath et. al. presented the important role of flotillins that regulate spatial organization of specific endosomal recyclings, specifically TCR sorting into Rab 11. The authors demonstrated their work by comparing with wild-type and flotillins knock-out cell lines and also developed a new method to visualize components in the recycling endosomes to the plasma membrane in real-time. Using the developed system, the authors dissected the regulation and importance of flotillin in Rab5 and Rab 11 mediated endosomal activities. Their novel system seems to be useful in the future but their findings are hard to understand and agree with from the current results. The developed model needs further characterization and their results contain a number of critical problems need to be accomplished. Also, they need serious modification of their paper.

We first would like to highlight that in light of the reviewers' comments and new data, we have substantially reorganised and streamlined the structure of the manuscript. This revised version is more compact and more focused on illustrating the mechanism by which flotillins contribute to sort TCR for recycling. For these reasons, the new manuscript no longer includes the data on phosphatidylserine and on transferrin recycling, which remain valid and relevant and will be part of a separate manuscript. We believe these changes, which have been made with the reviewers' comments in mind, truly make the manuscript better and clearer. The figure numbers mentioned in the response to the reviewers refer to the new version of the manuscript.

Major:

- Fig. 1: 1b, 1d: The authors demonstrate that without flotillins, TCR positive endosome around MTOC is impaired. The data demonstrate that accumulations at specific regions are impaired but does not demonstrate where the MTOC is exactly. To strengthen their point, they should observe with MTOC marker, e.g. using the centrosome marker.

We apologise for the confusion regarding what we wanted to illustrate in this figure. Our point was not to show an impaired recruitment to any specific structure. We only wanted to show that the distribution of Rab5 and Rab11a endosomes was disturbed by the absence of flotillins. We provide a quantification of the localisation of these endosomes (Fig. 4'f), which unquestionably illustrates that they are less aggregated at the centre of the cell, irrespectively of the position of the MTOC or any other organelle. To avoid confusion, we have removed any mention to the MTOC in the revised version of the manuscript.

Our data suggest that dynein-mediated transport might play a role in the endocytic sorting mechanism that relies on flotillins. And we are currently investigating if this is the case or not and if indeed the transport of endocytic vesicles towards the MTOC contributes to sorting TCR for recycling. However, these investigations fall beyond the scope of the present study and would not strengthen the core finding of the manuscript: that flotillins regulate sorting into and out of Rab5-positive endosomes to mediate Rab11a-dependent recycling.

- Fig. 2c: It is hard to observe the differences through the image. The color should be changed to black and white or inverted. Also, as demonstrated in 2c, it seems like mCherry tagged Flotillin1/2 was overexpressed in WT. I wonder why they did not use the KO cell line to minimize the secondary effect as possible. Numerous reports demonstrate the problems with overexpressing Flotillin. To further support their finding, they should rescue the Flotillin in KO cell to observe the fusion of TCR positive endosome.

We understand the reviewer's concern. However, we purposely did not use the FLotKO cell line in these experiments because the aim was to determine if fusing TCR-containing vesicles were positive for flotillin or for Rab11a. By using the FlotKO cell line to investigate flotillins, we would have used two different cell lines, one for flotillin (FLotKO) and another for Rab11a (WT), thereby introducing a bias in the comparison we aimed to perform and making the results incomparable to each other. We would like to further point out that if overexpression of flotillin would influence these experiments, it would be against our findings. Indeed, if flotillin was regulating the fusion of TCR-positive vesicles, its overexpression would probably results in more TCR-fusing vesicles positive for flotillin. Finally, we have to respectfully disagree with the reviewer. Studies reporting problems with overexpression of flotillin only show issues when only one of the two isoforms is expressed¹. This is the reason why we always overexpressed both isoforms. This is now more clearly stated in the text to alleviate this concern (page 10, lines 166-169).

- Fig. 2d: The result is confusing. On the previous report, the author's group demonstrated the importance of flotillins in TCR recycling. However, this result seems like in TCR recycling the independent of flotillin existence. The authors should clarify their result in more detail.

We have taken this comment into consideration in the revised version of our manuscript in order to avoid such confusion. Briefly, the aim of this study is to demonstrate that the role of flotillins in TCR recycling is to sort TCR into and through a Rab5-Rab11a recycling pathway. In these experiments we show that TCR is returned to the plasma membrane in Rab11a-positive vesicles, which do not contain flotillins. This is because flotillins act upstream, by promoting the sorting TCR into Rab11a endosomes. Once TCR is "loaded" into Rab11-positive vesicles, their travel and fusion to the plasma membrane does not require flotillins.

- Fig. 3: In this figure, the authors concluded the results that without flotillins, TCR is poorly sorted into Rab11 cycling vesicles. The authors should strengthen their point by demonstrating the fusion event with different controls. In Fig. 1, the localization of LACT-C2 (phosphatidylserine marker), Rab5 and Rab11 was affected but not with phosphoinositide PIP3, PIP4, Rab4 and Rab8. If their finding is specific for Rab11, than fusion events, especially with Rab4, should not be affected. Also try with Rab 11 mutants.

We thank the reviewer for the suggestion to investigate Rab4. Indeed, we were able to show that TCR incorporation into Rab4 compartments is completely independent of flotillins (Fig. 1b-d, page 5, lines 105-111). Additional data in the revised manuscript further show that Rab4 is not part of the endocytic pathway defined by flotillins (Fig. 4a, d, page 17, lines 240-245), in line with the absence of difference in the distribution of Rab4 endosomes between WT and FlotKO cells (Fig. 4e,f). It is therefore extremely unlikely that flotillins contribute to the fusion with the plasma membrane of Rab4-positive vesicles containing TCR.

We agree with the reviewer: investigating the exact function and role of Rab11a in T cells would be relevant and highly interesting. However, this is not the topic of this study. Indeed, the manuscript focuses on the function of flotillins and not on the role of Rab11. The use of Rab11 mutants would not help elucidating the contribution of flotillin in sorting TCR into a Rab5-Rab11a recycling pathway, as interfering with Rab11 function would be interfering downstream of the flotillin mediated sorting step. Accordingly, our data already show **a)** that flotillins sort TCR into Rab11a endosomes (Fig. 1h-j), **b)** that TCR contained in Rab11a endosomes is returned to the plasma membrane (Fig. 2f-h) and **c)** that TCR-positive vesicles fusing with the plasma membrane contain Rab11a (Fig. 3c-d). For these reasons, we do not think the usage of Rab11 mutants would strengthen our point.

- Since one of the strong points of this paper is the development of a novel method to visualize endosome fusion event, they should characterize this system in more detail by demonstrating how spatially the photo-activation can be reached to, e.g. range of spatial activation. Also the how long does it take for the activation of the molecule to be visualized.

We have extensively characterised how the photoactivation is localised in the cells with conventional and two-photon illumination (Fig. S2).

Photoactivation of fluorescent proteins is quasi instantaneous^{2,3} and required less than one frame in our experiments. The duration of the illumination with UV (time during which the fluorescent proteins are photoactivated) light is indicated in the figures (greyed area, for instance in Fig. 2c between 0 and 320 seconds).

- Fig. 4c-f: If the result is really specific to the existence of Flotillins, then other than Rab7 additional experiments with other controls are necessary (those that was not affected by Flot KO in Fig. 1). If Rab11 is the important factor, then the mutation of Rab11 should be added. Also why did Rab11a demonstrate higher TCR recycling event, while LACT-C2 and TCR increased the pixel in a similar manner? Rather than Student's t-test, the ANOVA with Tukey multiple comparison tests is more suitable to compare between different conditions

We show in new Fig. 1 that TCR is incorporated into Rab4-positive endosomes entirely independent of flotillins. It is therefore improbable that it would reach the plasma membrane from these compartments. Besides, the point of this experiment is only to show that TCR

recycles from flotillins- and Rab11a-positive endosomes, to illustrate that they are part of the same endocytic pathway. The conditions here are only compared to a baseline, defined by the amount of TCR returned from TCR positive compartments. Additional experiments with other markers of endosomes would only tell us if TCR recycles more or less from these endosomes relatively to the baseline. They would not provide additional information or confirmation regarding the fact that TCR is returned to the plasma membrane from flotillin and Rab11a-positive endosomes. Nevertheless, we have amended the manuscript to highlight the comparison to a baseline (page 9, lines 152-154) and to express more clearly the conclusions from these experiments (page 9, lines 174-176; 182-185).

Here again, we do not feel that Rab11a mutants would strengthen our point, as we are not claiming that Rab11a is an important factor in the sorting of TCR for recycling, but is a part of the machinery that returns it to the plasma membrane. Investigating how Rab11a regulates the transport and maybe the fusion with the plasma membrane of TCR-positive vesicles would be highly interesting, but this falls beyond the scope of the present manuscript.

As mentioned above, we have removed the phosphatidylserine data from the revised manuscript. Although they were valid and not in contradiction with the rest of the manuscript, we felt they were not bringing a sufficiently relevant contribution to explain how flotillins support the sorting of TCR for recycling.

An Anova test would indeed be more suitable if all the conditions would be compared together, or if all the tagged proteins would have been expressed in the same cells. In our case, the samples are always compared two by two and therefore we believe a t-test is more suitable. We nevertheless performed Anova tests on these data and it did not change the significance, if not making significant differences even more significant.

- Fig. 4g: "...TCR recycles from a flotillin-positive, Rab11a-positive..... but not flotillins" This sentence is hard to understand. What are the authors trying to say? It is also hard to understand the data from the legend and the main text. I guess the authors are trying to say is that photo-activation where flot1/2 was accumulated demonstrated recycling of TCR. But, no synchronized signal with Flot1/2 was observed but synchronize signal with Rab11 or LACT-C2 with TCR containing vesicle was observed. Again, the different control group is necessary to strengthen their points

We agree with the reviewer and have changed the sentence to make it less confusing (page 9, lines 183-185). The reviewer's comment was also one of the reasons why we modified the structure of the manuscript. We hope that the current order of the result paragraphs will dissipate the confusion regarding the role of flotillin in sorting TCR into Rab11a endosomes for a return to the plasma membrane. We have to disagree with the reviewer regarding further control groups. Here we characterised the propensity of vesicles positive for photoactivated TCR-PSCFP2 (i.e. initially contained inside the photoactivated compartments) to leave this compartment in vesicles that are also positive for the proteins defining the endosomes where they were photoactivated (flotillin1 and 2-mCherry or Rab11a-mCherry). In other words, we

see that when photoactivated in flotillin endosomes, TCR is not transported from this population of endosomes in vesicles positive for flotillin. By contrast, when photoactivated in Rab11a endosomes, TCR leaves this compartment in vesicles positive for Rab11a. This result is fully consistent with the data obtained in the TIRF vesicles fusion experiments (Fig. 3). Together, these data clearly illustrate that the role of flotillins is to get TCR into Rab11a endosomes and does not extend to the actual transport to the plasma membrane, which is achieved by Rab11a positive vesicles (Fig. 2 and 3). We have amended the text to make this point clearer (pages 14-15, lines 223-226).

- Fig. 5g: The photo-convergence fluorescent protein used in Fig. 4 and Fig. 5 are different. Therefore, it is not suitable to calculate how long TCR vesicle took for recycling, unless they quantify the photo-activated time of two fluorescent protein.

Following the reviewer's comment, we have amended the text make very clear that the time taken by TCR to recycle is only an estimation, based on the combination of two different assays (page 11, lines 186-88). Nevertheless, the time of photoconversion of photoactivatable proteins is negligible at the time scale at which these experiments are performed^{2,3} and is unlikely to have an influence on this estimation.

- Fig. 5h: The authors conclude that flotillin is critical to sort TCR into Rab5 to transfer into Rab11. Up to this point, the authors only focused on the activity of Rab 11. But this data demonstrate the importance of Rab 5 that consequently affected sorting into Rab 11. This data implies the previous results are not the important factor but consequent result from upstream events. Authors should focus more data with Rab5 in previous figures or else, the previous results seemed to be impractical.

We thank the reviewer for this constructive comment. Significantly more weight was given to Rab5 in the revised version of the manuscript. We also added additional experiments on Rab5, including how internalised flotillins interact with Rab5 endosomes (Fig. 4b,d, Movie 5) and the influence of light-induced aggregation of Rab5-positive endosomes on TCR recycling (Fig. 5f-i).

- In the chapter where Fig. S2 is present. Although this part is divided into a subgroup, there is no new information but just validating what was presented in previous studies with their technique. Also the cell they used was not presented. They mentioned that TCR reached phosphatidylserine-positive endosomes no longer than 7 seconds. Is this time mediated by Rab 5? Because on the previous chapter, they mentioned 6.5-7 minutes for TCR to return to PM. I am guessing 7 seconds is from when TCR is first internalized. Further information is needed

We have removed the data on phosphatidylserine from this manuscript, because, as rightfully mentioned in the reviewer's comment, they did not bring new information regarding the

mechanism we want to illustrate. Again, this does not mean the data were not valid nor interesting, but only that they will be more relevant within another manuscript.

- S3: In order to observe the spatial aggregation of Rab11, then the spatial light stimulation will be more adequate than whole cell stimulation.

The light-induced spatial aggregation of Rab11 using these optogenetic tools has already been characterised and published⁴. Therefore, the aim of this experiment was not to observe the spatial aggregation of Rab11a, but rather the consequence of it on TCR recycling. Additionally, spatially restricted aggregation of Rab11 could possibly be relevant when working with large adherent cells such as HeLa cells. But T cells are extremely small – 10 µm in diameter – and are almost entirely filled by their nucleus. This means there is only a very limited space for endosomes and would make the local clustering of only a part of Rab5 or Rab11a endosomes impossible. Furthermore, the aim of this experiment was to generate as much clustering of these endosomes as possible, therefore doing it in a restricted part of the cell would not have been adequate. We also would like to highlight that we have used these optogenetic tools as described in the original study⁴, where only whole cell stimulations were performed when investigating endocytic trafficking, even though they used larger cell types. Finally, locally aggregating Rab11 and at the same time visualising the recycling after two-photon photoactivation of TCR-PSCFP2 would simply not be feasible, as the 488 nm illumination required to detect PSCFP2 would trigger the aggregation of Rab11-Cry2 in the whole cell.

- Fig. 6b: there seemed to be no cluster formation other than the aggregation pattern due to Rab11 expression. To strengthen their point, they demonstrate the spatial light stimulation to generate aggregation on a specific location then perform their two-photon visualization modules. Also, they should compare with KO cell

The quantification presented in Fig. S2 clearly shows that Rab11a-positive structures – which are the only one we sought to cluster in this experiment – are clustered upon blue light exposure. This is also visible on the images in Fig. 5. We also have added a movie where the clustering of Rab11a-positive endosomes can clearly be observed (Movie 6). As mentioned above, the fact that endosomes are already compacted in the restricted cytoplasm of T cells should be taken into consideration when looking at these images. We also would like to encourage the reviewer to look at the data in the original publication⁴, which look similar to our data. We have discussed in the answer to the previous comment why local light-induced clustering of these endosomes in T cells would be irrelevant and, most importantly, impossible in the context of these experiments. Finally, doing these experiments in FlotKO cells would not provide further insights in the mechanism we intend to illustrate. The data shown in Fig. 5 show that Rab5 and Rab11a are indeed part of the endocytic pathway that recycles TCR. They also suggest that disturbing the spatial organisation of these endosomes –

which is also the consequence of knocking-out flotillins – impairs TCR return to the plasma membrane – which is also the consequence of the flotillin knock-out.

- Fig. 6: The authors concluded that spatial organization is important for recycling endosome. But Fig. 5h, the authors demonstrated that without flotillin, the sorting into Rab5 was also problematic. Therefore, they should also demonstrate the importance of flotillin with Rab5-mediated activity

Following the reviewer's suggestion, we have investigated how flotillins interact with Rab5 endosomes (Fig. 4b, d, Movie 5, page 17, lines 230-250) and how light-induced aggregation of Rab5 endosomes impair TCR return to the plasma membrane (Fig. 5 f-I, page 24, lines 315-318).

Fig. 7: It is hard to conclude that flotillin is universal regulators of endocytic cargo sorting since the authors are only looking at transferrin endocytosis activity. They should dissect other endocytosis activity

We agree with the reviewer, the word 'universal' was an overstatement. As mentioned at the beginning of our response, we have opted to remove the transferrin data from this manuscript. This allows its message to be clearer and more focused on TCR. The transferrin data and the phosphatidylserine data are part of a new manuscript currently in preparation.

- Also because Rab 5 is also perturbed, it is not reasonable to conclude that Rab 11 mediated activity is affected by flotillin. Rab11 sorting can be disrupted if Rab5 sorting has already defected.

We apologise for the confusion, as we never intended to show that Rab11-mediated activity is affected by flotillin, but rather that it is not (Fig. 3). The message we intend to convey in this study is in total agreement with the reviewer's comment: in absence of flotillins, less TCR is incorporated in Rab5 and less TCR is incorporated in Rab11a. The activity of Rab11, which in this instance is to bring TCR back to the plasma membrane, does not depend on flotillin. We have taken a great care to make this message clearer in the revised manuscript in order to avoid such confusion.

Minor:

Fig. 1: Figure numbering is poorly matched with the main text. As for Fig. 1, there are two b.

We have corrected the figure numbering.

Fig. 2a: Other than WT cell only, representative images or movie of reduced fusion events in the KO cells should be included, besides the quantified data.

We believe that showing the quantification is more adequate. It would be easy to pick a movie from one extreme or the other to illustrate our point, while showing the results from all the imaged cells in the quantification does not allow such a bias. Additionally, the fusion events themselves are not reduced; it is their frequency that is lower. The movies of KO cells look exactly the same than for WT, just that there are fewer fusion events taking place per minute in the KO cells. Figuring this out from the movies would require looking at them for quite some time, and to count the fusion events.

Fig. 3c, 3d: The fluorescence images are better than 2c but color changes might be better for better elucidation.

We consistently used the same colour scheme throughout the manuscript. We have chosen these colours because they are suitable for colour-blind people. Using a different colour scheme for one figure would be confusing.

Supplementary Fig. S1 is missing. S2 should be label into S1.

We have corrected the figure labelling.

References:

- 1 Frick, M. *et al.* Coassembly of flotillins induces formation of membrane microdomains, membrane curvature, and vesicle budding. *Curr. Biol.* **17**, 1151–6 (2007).
- 2 Chudakov, D. M., Lukyanov, S. & Lukyanov, K. A. Tracking intracellular protein movements using photoswitchable fluorescent proteins PS-CFP2 and Dendra2. *Nat. Protoc.* **2**, 2024–2032 (2007).
- 3 Subach, F. V *et al.* Photoactivatable mCherry for high-resolution two-color fluorescence microscopy. *Nat Methods* **6**, 153–159 (2009).
- 4 Nguyen, M. K. *et al.* Optogenetic oligomerization of Rab GTPases regulates intracellular membrane trafficking. *Nat. Chem. Biol.* **12**, 431–436 (2016).

Reviewer #3

- This paper provides new information on TCR sorting and recycling, and importantly, reveals a role of the flotillin proteins in cargo recycling (that was proposed earlier by the present group of authors and in earlier publications by other groups).

We first would like to highlight that in light of the reviewers' comments and new data, we have substantially reorganised and streamlined the structure of the manuscript. This revised version is more compact and more focused on illustrating the mechanism by which flotillins contribute to sort TCR for recycling. For these reasons, the new manuscript no longer includes the data on phosphatidylserine and on transferrin recycling, which remain valid and relevant and will be part of a separate manuscript. We believe these changes, which have been made with the reviewers' comments in mind, truly make the manuscript better and clearer. The figure numbers mentioned in the response to the reviewers refer to the new version of the manuscript.

- The central and novel observation in this paper is a disruption of the spatial organization of recycling endosomes in Jurkat T cells when flotillins are knocked out (ko). The authors show a reduction in TCR vesicle fusion in Flot ko cell lines compared to wt from average 10 events (?) in wt to average 5 events (?) in Flot ko per minute. Flot ko reduced the amount of TCR in Rab11 positive vesicles fusing with the membrane by about 40%. The authors conclude: `This implies that the TCR is poorly sorted into Rab11a recycling vesicles and consequently fails to be delivered to plasma membrane.`

Given that there is a reduction but not failure of TCR delivery to the PM, the above statement is a bit too strong!

We agree with the reviewer; our statement was too strong and we have rewritten this conclusion to more accurately describe our observations (page 15, lines 223-226).

- The authors use sophisticated methods which they established in their recent paper (Nat Comm, 2018) to visualize intracellular vesicle trafficking and cargo recycling. They succeed to activate specifically the fluorescence of TCR-PSCFP2 assembled near the centrosome, the position of the recycling compartment, thus avoiding confusion with signals from the PM.

Given that TCR is more widely dispersed in Flot ko cells, how can they be as precise in activating TCR at the centrosome in Flot ko cells?

This is a very relevant comment. TCR-positive endosomes show a more dispersed spatial organisation in FlotKO cells, and this could impede on the efficiency of the photoactivation of TCR-PSCFP2 within intracellular compartment in these cells. Following the reviewer's

comment, we have verified that the total amount of TCR-PSCFP2 being photoactivated was similar in WT and FlotKO cells (Fig. 2e), despite the difference in their spatial organisation.

Anyhow, with this method, they show a rapid recycling of the TCR back to the membrane (6-7min). In Flot ko cells, TCR was less efficiently transferred into Rab5 and Rab11 positive endosomes but was still occurring! This prompts the question what is the role of flotillins?

We hope that the way we have reorganised the manuscript illustrates better what we think the role of flotillins is! But to briefly answer the reviewer's question, we think our data are consistent with a role for flotillins in organising membranes, hence facilitating the transfer of TCR along the Rab5-Rab11a endocytic axis. However, without the compartmentalisation provided by flotillins, this transfer still happens, but less efficiently. Of note, if it still happens, the transfer of endocytosed TCR into Rab11a endosomes is quite significantly reduced (Fig. 1i).

The authors control for the specificity of this defect in TCR recycling by following transferrin recycling, and show that the transferrin present in Rab5 endosomes is reduced in Flot ko cells (70 % compared to 44% reduction of TCR). Transfer to Rab11 vesicles per se is not impaired but fusion with PM (40 and 20% in Flot k.o. versus 75% in WT).

Thus, there is a significant role of Flot1 and 2 in TCR and Tf recycling which, in case of Tf, is consistent with earlier results by other groups (Solis et al., 2013).

We agree, although the transferrin data are now part of a different manuscript. Nevertheless, we cite the study of Solis et al, as well as other studies reporting a role for flotillins in the context of Rab11-mediated recycling^{1,2} (page 17, line 230). However, we would like to mention that one of the main findings of this study is that flotillins play a role in sorting internalised TCR for recycling rather than in recycling *per se*.

The authors state that the spatial organization of the endosomal system is disrupted by Flot ko, and the images illustrating this effect are convincing. This is a nice result. However, it remains entirely unclear how flotillins would mechanistically contribute to establishment or maintenance of the spatial organization of the Rab5 and Rab11 positive endosomes. Flotillins are localized to the cytoplasmic face of the vesicle membranes. What is their idea how flotillins as microdomain forming membrane associated proteins regulate the spatial organization of the pre-early endosome and recycling compartment?

We would like to thank the reviewer for encouraging us to develop this point. As detailed in the revised version of the manuscript (page 18, lines 253-260; page 31, lines 381-387), we think that flotillin microdomains connect Rab5 and Rab11a endosomes to dynein motors, in the same way they do on the membrane of phagosomes³. Indeed, we observed that Rab5 and Rab11a endosomes were more at the periphery/less at the centre in the flotillin knock-out T cells, which is consistent with the direction of dynein-powered transport. Although this

hypothesis is consistent with our data and with the literature, it remains a hypothesis, which we aim to strengthen in the coming years.

Flotillins are abundant at the Rab11 recycling compartment (Gagescu et al 2000; Solomon et al., 2002) and less so at Rab5 positive endosomes.

We agree with the reviewer, there is very little evidence of a connection between flotillins and Rab5 in the literature. However, a recent study has reported transient interactions⁴, very similar to those that we observed in Fig. 4b and Movie 5, between Rab5 and flotillins (in the figure 7A and S4D of the referenced paper). Our data also show that Rab5-flotillin interactions are very transient and short lasting, and this could be the reason why they have not been much documented before. Of note, a mass spectrometry analysis revealed that flotillins, Rab5 and Rab11 were shown to be part of the same endosomal fraction in neurones⁵

The authors show that flotillin vesicles are positive for phosphatidyl serine (PS) and Rab11a but are they Rab5 positive as is shown on their scheme in Fig.8? Should the arrow from the putative pre-early endosome not rather point to the transit from Rab5 to Rab11 (bypass Rab5)? This needs to be experimentally clarified. What exactly do they mean by pre-early endosome? This aspect (see also Discussion lines 357 to 363) remains vague and controversial (...sorting prior to entry into Rab5...).

The question of the presence of flotillin in Rab5-positive membrane seems indeed to be a complex one. We have taken the reviewer's comment into consideration and tried to experimentally clarify the connection of flotillin with Rab5 (Fig. 4b,d, Movie 5). Altogether, our data show that flotillins are required for internalised TCR to get into Rab5 endosomes (Fig. 1f), to exit Rab5 endosomes (Fig. 1g) and to enter Rab11a endosomes (Fig. 1i,j). But crucially, they further show that endocytosed flotillins are found, although transiently and with a low frequency, in Rab5 endosomes (new data, Fig. 4b,d, Movie 5). In the model that we propose at the end of the discussion (page 32, lines 389-404), flotillins define a small minority of Rab5-positive membranes, for a short period of time, somehow facilitating the passage of TCR through Rab5 endosomes and to Rab11a endosomes, where these "flotillin-positive membrane" finally end up. This is consistent with all our data and would explain why the connection of flotillins with Rab11 is well-established whereas only one study reports a link with Rab5.

That flotillins might mediate the interaction with cytoskeletal elements is discussed towards the end of the manuscript but not shown in the scheme (Fig.8). It seems to me that the reader needs more information as to how the spatial order of endosomes is perturbed in Flot knock out cells and how cargo sorting is affected.

We did not include the connection to the cytoskeleton in the scheme because at this stage it remains a hypothesis in our model. However, we fully agree with the reviewer regarding the fact that readers need more information on the mechanism through which flotillins support the spatial organisation of endosomes. This is the reason why we have included a proposed model at the end of the discussion, where we discuss the potential role of membrane domains and dynein (page 32, lines 392-399).

The authors use Jurkat cells in which flotillin-1 and 2 were knocked out. They concentrate on the analysis of the TCR recycling pathway. They do not comment on which other aspects the Jurkat T cells may become abnormal (or remain normal ?) when flotillins are missing.

The reason why did not comment on other consequences of flotillin knock out in Jurkats cells is because this was part of a previously published study⁶. Briefly, we showed that flotillin-mediated TCR recycling is essential for T cell activation, supporting TCR nanoscale organization and signalling.

From work by other groups flotillins are known to form a preformed cap opposite from the MTOC/centrosome and recycling compartment. This is not included in their scheme of Fig 8, and is mentioned no-where in the text.

A cap at the plasma membrane defined by flotillin has indeed been described in spherical T cells, non-activated or activated with PMA. We believe this cap is more related to T cell polarity and formation of a uropod in the context of cell migration. We do not think this relates to the mechanism described by our study and this is why we have not mentioned it. Nevertheless, we mentioned the ability of flotillin to organise T cell plasma membrane and made sure to cite the relevant studies.

This is to say that probably additional aspects of disorder might occur in the absence of flotillins. And the absence of flotillins from the PM might have severe effects.

Some disorganisation of the plasma membrane in absence of flotillins would be consistent with our model, as it could impair the transfer of TCR from the plasma membrane to Rab5 endosomes. However, we do not believe the plasma member is severely disorganised in the FlotKO T cells., Following the reviewer's concern, we have investigated the plasma membrane of WT and FlotKO T cells using electron microscopy. We could not observe any detectable differences in the organisation or morphology of the plasma membrane. To demonstrate this point, we have included a number of high-resolution EM images in a new figure with magnified regions of the plasma membrane included (Fig. S3).

The role of flotillins in sorting and recycling has been shown earlier by other investigators and needs to be cited appropriately. Their presence near the centrosome has also been demonstrated earlier as well as the interaction of flotillins with the cytoskeleton. Proper citations are necessary. That flotillins are also known under reggies and that knock out mice are existing needs to be mentioned as well.

We agree with the reviewer's comment and have tried to cite the relevant literature more accurately in the revised version of the manuscript. We also referred to the fact that flotillins are known as reggies (page 3, line 69).

Altogether, the manuscript contains highly interesting results concerning TCR recycling and concerning the role of flotillins in intracellular trafficking. But it seems that more work is needed towards the mechanism of flotillin functions.

We hope the additional experiments (Fig. 4a-d) and the model proposed in the revised version (page 32, lines 389-404) describe more satisfactorily the mechanism of flotillin functions in T cells.

References:

1. Bodrikov, V., Pauschert, A., Kochlamazashvili, G. & Stuermer, C. a O. Reggie-1 and reggie-2 (flotillins) participate in Rab11a-dependent cargo trafficking, spine synapse formation and LTP-related AMPA receptor (GluA1) surface exposure in mouse hippocampal neurons. *Exp. Neurol.* **289**, 31–45 (2017).
2. Hülbusch, N., Solis, G. P., Katanaev, V. L. & Stuermer, C. A. O. Reggie-1/Flotillin-2 regulates integrin trafficking and focal adhesion turnover via Rab11a. *Eur. J. Cell Biol.* **94**, 531–45 (2015).
3. Rai, A. *et al.* Dynein Clusters into Lipid Microdomains on Phagosomes to Drive Rapid Transport toward Lysosomes. *Cell* **164**, 722–34 (2016).
4. Planchon, D. *et al.* MT1-MMP targeting to endolysosomes is mediated by upregulation of flotillins. *J. Cell Sci.* **131**, jcs218925 (2018).
5. Kokotos, A. C., Peltier, J., Davenport, E. C., Trost, M. & Cousin, M. A. Activity-dependent bulk endocytosis proteome reveals a key presynaptic role for the monomeric GTPase Rab11. *Proc. Natl. Acad. Sci.* **115**, E10177–E10186 (2018).
6. Compeer, E. B. *et al.* A mobile endocytic network connects clathrin-independent receptor endocytosis to recycling and promotes T cell activation. *Nat. Commun.* **9**, 1597 (2018)

REVIEWERS' COMMENTS:

Reviewer #2 (Remarks to the Author):

The authors have addressed most of my comments. The revised manuscript is now much refined. I recommend that the revised manuscript be suitable for publication in Nature Communications.

Editorial Note: We have asked a new Reviewer (Reviewer #4) to comment on the responses to Reviewer #3's previous concerns.

“Flotillins promote T cell receptor sorting through a fast Rab5-Rab11 endocytic recycling axis” (NCOMMS-18-23454-T)

Remarks to comments of Reviewer 3 and to author response are shown in **bold!**

- The authors use sophisticated methods which they established in their recent paper (Nat Comm, 2018) to visualize intracellular vesicle trafficking and cargo recycling. They succeed to activate specifically the fluorescence of TCR-PSCFP2 assembled near the centrosome, the position of the recycling compartment, thus avoiding confusion with signals from the PM.

Given that TCR is more widely dispersed in Flot ko cells, how can they be as precise in activating TCR at the centrosome in Flot ko cells?

This is a very relevant comment. TCR-positive endosomes show a more dispersed spatial organisation in FlotKO cells, and this could impede on the efficiency of the photoactivation of TCR-PSCFP2 within intracellular compartment in these cells. Following the reviewer's comment, we have verified that the total amount of TCR-PSCFP2 being photoactivated was similar in WT and FlotKO cells (Fig. 2e), despite the difference in their spatial organisation.

This comment has been addressed properly!

Anyhow, with this method, they show a rapid recycling of the TCR back to the membrane (6-7min). In Flot ko cells, TCR was less efficiently transferred into Rab5 and Rab11 positive endosomes but was still occurring! This prompts the question what is the role of flotillins?

We hope that the way we have reorganised the manuscript illustrates better what we think the role of flotillins is! But to briefly answer the reviewer's question, we think our data are consistent with a role for flotillins in organising membranes, hence facilitating the transfer of TCR along the Rab5-Rab11a endocytic axis. However, without the compartmentalisation provided by flotillins, this transfer still happens, but less efficiently. Of note, if it still happens, the transfer of endocytosed TCR into Rab11a endosomes is quite significantly reduced (Fig. 1i).

The explanation that the transfer is impaired but not completely inhibited is plausible. However, this remains the weak point of the manuscript: what is the direct molecular mechanism of flotillin action during this sorting event?

The authors control for the specificity of this defect in TCR recycling by following transferrin recycling, and show that the transferrin present in Rab5 endosomes is reduced in Flot ko cells (70 % compared to 44% reduction of TCR). Transfer to Rab11 vesicles per se is not impaired but fusion with PM (40 and 20% in Flot k.o. versus 75% in WT). Thus, there is a significant role of Flot1 and 2 in TCR and Tf recycling which, in case of Tf, is consistent with earlier results by other groups (Solis et al., 2013).

We agree, although the transferrin data are now part of a different manuscript. Nevertheless, we cite the study of Solis et al, as well as other studies reporting a role for flotillins in the

context of Rab11-mediated recycling^{1,2} (page 17, line 230). However, we would like to

mention that one of the main findings of this study is that flotillins play a role in sorting internalised TCR for recycling rather than in recycling *per se*.

Transferrin data have been removed, so this point is not relevant anymore!

The authors state that the spatial organization of the endosomal system is disrupted by Flot ko, and the images illustrating this effect are convincing. This is a nice result. However, it remains entirely unclear how flotillins would mechanistically contribute to establishment or maintenance of the spatial organization of the Rab5 and Rab11 positive endosomes. Flotillins are localized to the cytoplasmic face of the vesicle membranes. What is their idea how flotillins as microdomain forming membrane associated proteins regulate the spatial organization of the pre-early endosome and recycling compartment?

We would like to thank the reviewer for encouraging us to develop this point. As detailed in the revised version of the manuscript (page 18, lines 253-260; page 31, lines 381-387), we think that flotillin microdomains connect Rab5 and Rab11a endosomes to dynein motors, in the same way they do on the membrane of phagosomes³. Indeed, we observed that Rab5 and Rab11a endosomes were more at the periphery/less at the centre in the flotillin knock-out T cells, which is consistent with the direction of dynein-powered transport. Although this hypothesis is consistent with our data and with the literature, it remains a hypothesis, which we aim to strengthen in the coming years.

The authors honestly admit that they do not know what the mechanism is and aim at clarifying the details by further studies. It is up to the Editor to decide if this is sufficient or if some kind of a mechanism is required!

The authors show that flotillin vesicles are positive for phosphatidyl serine (PS) and Rab11a but are they Rab5 positive as is shown on their scheme in Fig.8? Should the arrow from the putative pre-early endosome not rather point to the transit from Rab5 to Rab11 (bypass Rab5)? This needs to be experimentally clarified. What exactly do they mean by pre-early endosome? This aspect (see also Discussion lines 357 to 363) remains vague and controversial (...sorting prior to entry into Rab5...).

The question of the presence of flotillin in Rab5-positive membrane seems indeed to be a complex one. We have taken the reviewer's comment into consideration and tried to experimentally clarify the connection of flotillin with Rab5 (Fig. 4b,d, Movie 5). Altogether, our data show that flotillins are required for internalised TCR to get into Rab5 endosomes (Fig. 1f), to exit Rab5 endosomes (Fig. 1g) and to enter Rab11a endosomes (Fig. 1i,j). But crucially, they further show that endocytosed flotillins are found, although transiently and with a low frequency, in Rab5 endosomes (new data, Fig. 4b,d, Movie 5). In the model that we propose at the end of the discussion (page 32, lines 389-404), flotillins define a small minority of Rab5-positive membranes, for a short period of time, somehow facilitating the passage of TCR through Rab5 endosomes and to Rab11a endosomes, where these "flotillin-positive membrane" finally end up. This is consistent with all our data and would explain why the connection of flotillins with Rab11 is well-established whereas only one study reports a link with Rab5.

I deeply agree with the authors about this point: flotillins seem to be some kind of omnipotent all-rounders that are present in many compartments and seem to regulate various transport steps. So far, nobody has been able to clarify how this exactly works and what makes flotillins so promiscuous but vital for vesicular trafficking.

That flotillins might mediate the interaction with cytoskeletal elements is discussed towards the end of the manuscript but not shown in the scheme (Fig.8). It seems to me that the reader needs more information as to how the spatial order of endosomes is perturbed in Flot knock out cells and how cargo sorting is affected.

We did not include the connection to the cytoskeleton in the scheme because at this stage it remains a hypothesis in our model. However, we fully agree with the reviewer regarding the fact that readers need more information on the mechanism through which flotillins support the spatial organisation of endosomes. This is the reason why we have included a proposed model at the end of the discussion, where we discuss the potential role of membrane domains and dynein (page 32, lines 392-399).

This response is merely an attempt to avoid addressing this experimentally. However, I do not think that such studies are required in this manuscript!

The authors use Jurkat cells in which flotillin-1 and 2 were knocked out. They concentrate on the analysis of the TCR recycling pathway. They do not comment on which other aspects the Jurkat T cells may become abnormal (or remain normal ?) when flotillins are missing.

The reason why did not comment on other consequences of flotillin knock out in Jurkats cells is because this was part of a previously published study⁶. Briefly, we showed that flotillin-mediated TCR recycling is essential for T cell activation, supporting TCR nanoscale organization and signalling.

Response makes fully sense!

From work by other groups flotillins are known to form a preformed cap opposite from the MTOC/centrosome and recycling compartment. This is not included in their scheme of Fig 8, and is mentioned no-where in the text.

A cap at the plasma membrane defined by flotillin has indeed been described in spherical T cells, non-activated or activated with PMA. We believe this cap is more related to T cell polarity and formation of a uropod in the context of cell migration. We do not think this relates to the mechanism described by our study and this is why we have not mentioned it. Nevertheless, we mentioned the ability of flotillin to organise T cell plasma membrane and made sure to cite the relevant studies.

I also agree on this!

This is to say that probably additional aspects of disorder might occur in the absence of flotillins. And the absence of flotillins from the PM might have severe effects.

Some disorganisation of the plasma membrane in absence of flotillins would be consistent with our model, as it could impair the transfer of TCR from the plasma membrane to Rab5 endosomes. However, we do not believe the plasma membrane is severely disorganised in the FlotKO T cells. Following the reviewer's concern, we have investigated the plasma membrane of WT and FlotKO T cells using electron microscopy. We could not observe any detectable differences in the organisation or morphology of the plasma membrane. To demonstrate this point, we have included a number of high-resolution EM images in a new figure with magnified regions of the plasma membrane included (Fig. S3).

I think it is very much likely that there are changes in the organization of the PM upon flotillin KO. However, the authors are right about the potential effect on TCR trafficking. I agree with this response!

The role of flotillins in sorting and recycling has been shown earlier by other investigators and needs to be cited appropriately. Their presence near the centrosome has also been demonstrated earlier as well as the interaction of flotillins with the cytoskeleton. Proper citations are necessary. That flotillins are also known under reggies and that knock out mice are existing needs to be mentioned as well.

We agree with the reviewer's comment and have tried to cite the relevant literature more accurately in the revised version of the manuscript. We also referred to the fact that flotillins are known as reggies (page 3, line 69).

References are now OK!

Altogether, the manuscript contains highly interesting results concerning TCR recycling and concerning the role of flotillins in intracellular trafficking. But it seems that more work is needed towards the mechanism of flotillin functions.

We hope the additional experiments (Fig. 4a-d) and the model proposed in the revised version (page 32, lines 389-404) describe more satisfactorily the mechanism of flotillin functions in T cells.

I agree with the reviewer: the molecular mechanism is still missing! Very interesting story despite this point.